# Rescue of neuropsychiatric phenotypes in a mouse model of 16p11.2 duplication syndrome by genetic correction of an epilepsy network hub

Marc P. Forrest [1,2], Marc Dos Santos [1,2], Nicolas H. Piguel[1,2], Yi-Zhi Wang[3], Nicole A. Hawkins[4], Vikram A. Bagchi[1,2], Leonardo E. Dionisio[1,2], Sehyoun Yoon[1,2], Dina Simkin [3,4], Maria Dolores Martin-de-Saavedra [1,2,7], Ruoqi Gao[1,2], Katherine E. Horan[1,2], Alfred L. George Jr. [2,4], Mark S. LeDoux[5,6], Jennifer A. Kearney[2,4], Jeffrey N. Savas [2,3] & Peter Penzes[1,2] ✉

Neuropsychiatric disorders (NPDs) are frequently co-morbid with epilepsy, but the biological basis of shared risk remains poorly understood. The 16p11.2 duplication is a copy number variant that confers risk for diverse NPDs including autism spectrum disorder, schizophrenia, intellectual disability and epilepsy. We used a mouse model of the 16p11.2 duplication (16p11.2$^{dup/+}$) to uncover molecular and circuit properties associated with this broad phenotypic spectrum, and examined genes within the locus capable of phenotype reversal. Quantitative proteomics revealed alterations to synaptic networks and products of NPD risk genes. We identified an epilepsy-associated subnetwork that was dysregulated in 16p11.2$^{dup/+}$ mice and altered in brain tissue from individuals with NPDs. Cortical circuits from 16p11.2$^{dup/+}$ mice exhibited hypersynchronous activity and enhanced network glutamate release, which increased susceptibility to seizures. Using gene co-expression and interactome analysis, we show that PRRT2 is a major hub in the epilepsy subnetwork. Remarkably, correcting *Prrt2* copy number rescued aberrant circuit properties, seizure susceptibility and social deficits in 16p11.2$^{dup/+}$ mice. We show that proteomics and network biology can identify important disease hubs in multigenic disorders, and reveal mechanisms relevant to the complex symptomatology of 16p11.2 duplication carriers.

Neuropsychiatric disorders are leading causes of global disability, but the underlying disease mechanisms are still poorly understood[1,2]. NPDs such as autism spectrum disorder (ASD), schizophrenia (SZ), intellectual disability (ID) are known to share behavioral symptoms and co-morbidities[3–5]. Notably, all these conditions have a strong epidemiological association with epilepsy, indicating the potential for shared risk mechanisms[6–9]. Epilepsy is frequently comorbid in individuals with

ASD, with (22%) or without (8%) ID, while ASDs occurs in 4–20% of children with epilepsy[7,10,11]. Similarly, there is an elevated prevalence and risk for schizophrenia in epilepsy estimated at 5–10% of individuals[9,12,13]. Allied with epidemiological studies, human genetic studies have demonstrated an overlap between risk factors for epilepsy and those of common psychiatric disorders[14]. Thus, epilepsy shares a genetic basis and co-occurs widely with NPDs, but

the biological substrates of this co-morbidity remain poorly understood. Expanding our understanding of the molecular and circuit bases of epilepsy co-morbidity could provide insight into NPD pathophysiology and renewed treatment strategies that cross diagnostic boundaries.

Copy number variants (CNVs) have emerged as highly penetrant risk variants that increase risk for multiple disorders, offering powerful models to better understand the convergent etiology of NPDs[15]. Among these, the proximal 16p11.2 duplication (BP4–BP5), is remarkable for its extensive pleiotropy, conferring risk for a wide variety of NPDs, including epilepsy. The CNV encompasses approximately 600 kb on human chromosome 16, a region encoding 30 genes (27 protein coding), many of which have poorly defined functions in the brain (Supplementary Fig. 1a). Meta-analyses have identified this CNV as one of the most statistically robust variants associated with SZ and ASD (odd ratios 8–14), confirming initial studies[16–19]. Furthermore, the 16p11.2 duplication confers risk for intellectual disability, bipolar disorder, attention deficit hyperactivity disorder (ADHD), depression, and Rolandic epilepsy[20–24]. Clinically ascertained cohorts have shown that the most prevalent conditions include: ASD, ID, ADHD, psychosis and seizures[25–27]. These complex neuropsychiatric phenotypes have led to the delineation of a 16p11.2 duplication syndrome [MIM: 614671]. Mouse models of the syntenic 16p11.2 duplication region have been generated (16p11.2$^{dup/+}$ mice), and have variable impairments in locomotor activity, working memory, repetitive and social behaviors, but clear disease mechanisms, and genotype-phenotype relationships, are still lacking[28–31].

Advancements in proteomic techniques and sample preparation protocols, have transformed the large-scale study of proteins; building blocks of neuronal circuits[32]. Although RNA-sequencing is widely used in neuroscience, proteomics offers a new perspective to study altered molecular pathways, which cannot always be predicted by transcriptomics[33]. Importantly, proteomics can offer increased resolution on subcellular compartments such as cellular membranes, key sites of cell-cell communication, and uncover critical protein:protein interaction networks. In combination with network biology, proteomics can be used to isolate important disease processes, identify core networks and critical hub proteins that may mediate pathogenesis[34]. Thus, proteomics is promising approach to discover novel molecular and circuit mechanisms associated with complex genetic disorders, and uncover hub proteins capable of reversing disease biology.

Here, we use a systems biology approach, combining proteomics with human genetic datasets, to uncover disease networks, and critical hub proteins, that are disrupted in the membrane proteome of the 16p11.2$^{dup/+}$ mouse model. We identify a disrupted network of epilepsy and synaptic gene-products in the 16p11.2$^{dup/+}$ mouse model, which is concomitant with hypersynchronous circuitry, enhanced network glutamate release and increased seizure susceptibility. Bioinformatic prediction and protein:protein interaction analysis further identified PRRT2, a protein encoded within the 16p11.2 region, as a highly connected hub within the epilepsy subnetwork disrupted in 16p11.2$^{dup/+}$ mice. Remarkably, genetic correction of subnetwork hub PRRT2 rectified aberrant circuit function, improved seizure susceptibility, and rescued social deficits in 16p11.2$^{dup/+}$ mice. Our work reveals that proteomics and network biology can be used to dissect disease mechanisms in complex CNVs, revealing molecular and circuit mechanisms that could underlie distinct neurological and psychiatric symptoms.

## Results

### Membrane proteomics uncovers disrupted synaptic and epilepsy-associated networks in 16p11.2$^{dup/+}$ mice

The 16p11.2 duplication alters the structure of multiple brain regions in mice and humans[28,35]. Here, we focused on studying proteomic dysfunction in the neocortex, a brain region that exhibits molecular and

structural abnormalities in several NPDs[36,37]. We performed a quantitative proteomic analysis of neocortical membranes because these sub-compartments contain a diverse array of transporters, cell adhesion molecules, receptors, ion channels and enzymes, which integrate many aspects of neuronal physiology, and could provide insights into cellular and circuit function. In addition, the composition of the membrane proteome may reveal novel disease mechanisms that cannot be captured by transcriptomics or whole-tissue proteomics[38]. We first validated the altered protein expression in the 16p11.2$^{dup/+}$ mouse model by Western blotting (Supplementary Fig. 1b) and then fractionated the neocortex of 16p11.2$^{dup/+}$ mice to obtain membrane fractions. We employed a stable isotope labeling in mammals (SILAM)-based quantitative proteomic technique using $^{15}$N-labeled mouse brains as internal standards to compare 16p11.2$^{dup/+}$ and wild-type membrane fractions, which were then analyzed by liquid chromatography-tandem mass spectrometry (Fig. 1a). We identified ~4400 proteins representing diverse membranous compartments including vesicles, dendrites, axons and synapses (Supplementary Fig. 1c). Our approach revealed a widespread proteomic dysregulation of neocortical membranes in 16p11.2$^{dup/+}$ mice, including 659 significantly upregulated and 1024 significantly downregulated proteins (Fig. 1b) (Supplementary Data 1). Importantly, PRRT2 (Z-ratio = 115.6) and SEZ6L2 (Z-ratio = 23.3), the two brain-expressed membrane proteins encoded in the 16p11.2 region were identified as upregulated in membrane fractions. Globally, membrane proteome changes were enriched for genes products that preferentially express in excitatory neurons versus inhibitory neurons or other cell types (Supplementary Fig. 1d). Using gene ontology (GO) analysis, we found that upregulated proteins were significantly enriched for biological pathways regulating multiple aspects of circuit activity, including synaptic signaling, ion transport, exocytosis and action potential generation (Supplementary Data 2) (Fig. 1c). Conversely, downregulated proteins had only modest enrichment for very few pathways. A more detailed analysis of synaptic ontologies using SynGO revealed that upregulated proteins were particularly enriched for presynaptic processes including synaptic vesicle cycle and exocytosis (Supplementary Fig. 1e, f) (Supplementary Data 3).

Alterations in membrane protein abundance could be caused by several mechanisms including dysregulation of transcription, proteostasis and/or protein trafficking. To gain insight into the molecular mechanisms involved, we compared our proteomic data to two published datasets that used RNA-seq to profile gene expression changes in the whole neocortex or prefrontal cortex of 16p11.2$^{dup/+}$ mice[30,39] (Supplementary Fig. 2a–c). In the whole neocortex dataset, we found 41 genes that overlapped with the membrane proteome changes (Supplementary Fig. 2a). Enrichment analysis revealed this overlap was not statistically significant, both with, or without 16p11.2 genes included in the analysis (Supplementary Fig. 2d, e). We also considered genes/proteins in the whole cortex dataset that were altered in the same direction (i.e. up in RNA-seq vs up in membrane proteome or down in RNA-seq and down in membrane proteome), and again found no statistically significant enrichment (Supplementary Fig. 2d, e). We next performed a correlation analysis with gene/proteins in the overlap (n = 41) and found that they were significantly correlated (p = 0.0378), albeit with a poor linear relationship (R$^2$ = 0.022) (Supplementary Fig. 2f). However, the significance of this correlation disappeared (p = 0.1638) when genes in the 16p11.2 locus were removed, indicating that the correlation was dependent on cis effects of the 16p11.2 region (Supplementary Fig. 2g). In the prefrontal cortex dataset, we found 53 genes that overlapped with the membrane proteome changes, which represented an enrichment more than was expected by chance (p = 0.0036) (Supplementary Fig. 2b, h). This enrichment persisted when we removed the 16p11.2 locus genes (p = 0.0024) (Supplementary Fig. 2i). However, when considering genes/proteins that were altered in the same direction, we found no statically significant

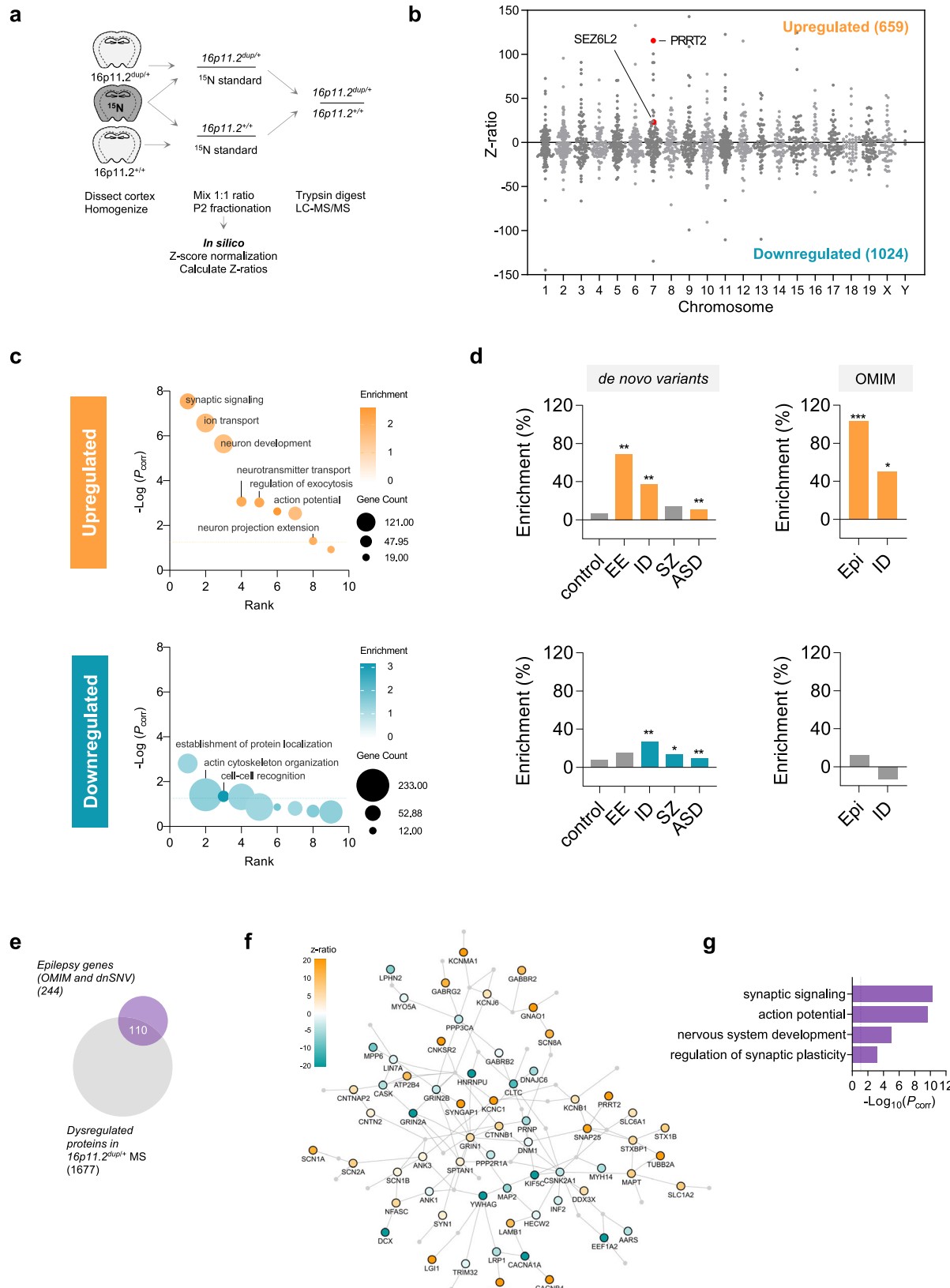

enrichment, which was corroborated by the lack of correlation between the two data sets (Supplementary Fig. 2j, k). Together these data indicate a lack of correlation between gene expression and membrane proteomic changes for genes/proteins located outside of the 16p11.2 locus, and suggest that membrane changes are likely not due to global transcriptional effects.

Given that human 16p11.2$^{dup/+}$ carriers exhibit a broad phenotypic spectrum, we set out to determine if there were molecular networks associated to NPDs represented within the proteomic data. To this end, we used gene set analysis to determine if the dysregulated proteins contained a burden of disease associated gene products identified by de novo exome sequencing studies of NPDs. We used exome

**Fig. 1 | Quantitative membrane proteomics reveals disrupted synaptic and epilepsy-associated protein networks in the 16p11.2$^{dup/+}$ mouse model.**
**a** Procedure for SILAM-based quantitative proteomic profiling of cortical membranes from 16p11.2$^{+/+}$ (*n* = 5 mice) and 16p11.2$^{dup/+}$ (*n* = 5 mice) mice using $^{15}$N-enriched brains as internal standards. **b** Graphical display of proteomic dysregulation in cortical membranes of 16p11.2$^{dup/+}$ mice. Each protein is plotted by Z-ratio (from −150 to +150 for clarity) and ordered by chromosome to visualize the extent of dysregulation across the proteome. We identified 659 upregulated proteins (Z-ratio >1.96) and 1024 downregulated proteins (Z-ratio < −1.96) (full dataset available in Supplementary Data 1). Membrane proteins PRRT2 and SEZ6L2 within the syntenic 16p11.2 region (murine 7q4) are indicated. **c** Gene ontology (GO) analysis reveals an upregulation of synaptic signaling and ion transport proteins. Representative terms from the top GO clusters are presented **d** Molecular risk profile of the 16p11.2$^{dup/+}$ mouse model. Comparison of dysregulated proteins from the proteomic profiling to gene sets of de novo variants identified through large-scale exome sequencing studies or the OMIM database. Products of genes

impacted by de novo variants in epilepsy are highly over-represented in the upregulated protein dataset. Data shows percentage enrichment and *p*-values from a two-sided hypergeometric test (Upregulated: control, *p* = 0.2954; EE, *p* = 0.0011; ID, *p* = 0.0059; SZ, *p* = 0.0660; ASD, *p* = 0.0094; Epi (OMIM), *p* = 0.0001; ID (OMIM), *p* = 0.0374. Downregulated: control, *p* = 0.1687; EE, *p* = 0.1922; ID, *p* = 0.0073; SZ, *p* = 0.0235; ASD, *p* = 0.0028; Epi (OMIM), *p* = 0.2965; ID (OMIM), *p* = 0.2922).
**e** Schematic showing the 110 dysregulated proteins associated with epilepsy used to seed the protein interaction network. **f** The epilepsy-associated protein interaction network (Epilepsy subnetwork) discovered in the 16p11.2$^{dup/+}$ mouse model. Nodes are colored by Z-ratio. **g** GO analysis of the epilepsy subnetwork reveals a strong enrichment of synaptic proteins (one-sided Fisher's Exact test with Benjamini-Hochberg correction). Abbreviations: $^{*}p < 0.05$, $^{**}p < 0.01$, $^{***}p < 0.001$, EE epileptic encephalopathy, Epi epilepsy, ID intellectual disability, SZ schizophrenia, ASD autism spectrum disorder, RE Rolandic epilepsy, BD bipolar disorder, $P_{corr}$, Benjamini-Hochberg corrected *p*-value.

sequencing data because these implicate single genes in disease, and can therefore be employed to identify disease-specific protein networks with defined biological functions[40]. To uncover these molecular networks, we compared sets of dysregulated proteins identified in our proteomic study to disease-specific gene sets; generating a proteomic disease profile for cortical membranes in the 16p11.2$^{dup/+}$ mouse. Importantly, we used all identified membrane proteins within our dataset as a background for these analyses, removing any bias that may arise from brain expression or membrane localization. We found an increased burden of risk factors associated with many 16p11.2$^{dup/+}$-related NPDs in our dataset (Supplementary Fig. 3a and Fig. 1d). These data are reminiscent of the diverse clinical spectrum observed in CNV carriers and suggest that subnetworks relevant to each disease may be dysregulated in 16p11.2$^{dup/+}$ mice. The most enriched disease gene set was associated with epilepsy risk factors, followed by ID, SZ and ASD gene sets (Supplementary Fig. 3a). Most of the epilepsy gene set was contained in the upregulated proteins (+69%, *p* = 1.07 × 10$^{-3}$), whilst SZ gene sets were enriched among the downregulated proteins (+14%, *p* = 2.35 × 10$^{-2}$). ID and ASD gene sets were enriched in both up and downregulated datasets. Importantly, control datasets of de novo single nucleotide variants (dnSNVs) identified in non-affected individuals were not enriched in any proteomic dataset, demonstrating a selective enrichment for NPD risk factors (Supplementary Fig. 3a, Fig. 1d). To confirm the enrichment for epilepsy risk factors in the proteomic dataset we used an independent gene set derived from a catalog of Mendelian disease genes (OMIM) associated with epilepsy or ID. This analysis revealed an even stronger enrichment for epilepsy (+103%, *p* = 1.11 × 10$^{-2}$), suggesting a robust association between the altered proteins in cortical membranes of 16p11.2$^{dup/+}$ mice and biological pathways related to epilepsy.

**An epilepsy subnetwork is altered in neuropsychiatric disorders**
We next aimed to determine if epilepsy-related proteins in our data set were functionally independent or formed part of a larger protein-protein interaction (PPI) network with enriched biological functions. We seeded a network with 110 proteins disrupted in the 16p11.2$^{dup/+}$ mouse model and genetically associated with epilepsy (Fig. 1e). By matching with a PPI database[41], we found that the majority of epilepsy-associated proteins were physically interconnected (100 nodes, 137 edges), potentially forming a large functional module (Fig. 1f and Supplementary Fig. 3b). GO analysis further showed that the epilepsy-associated network was highly enriched for synaptic ontologies (Fig. 1g and Supplementary Fig. 3c) (Supplementary Data 4).

To determine the broader relevance of this network, we aimed to determine whether proteins disrupted in the epilepsy network and membrane proteome of 16p11.2$^{dup/+}$ mice were also altered in NPD mouse models, and human post-mortem brain tissue in ASD and SZ (Supplementary Fig. 3d). To this end, we overlapped lists of

proteins dysregulated in the 16p11.2$^{dup/+}$ mouse model with dysregulated proteins from mouse or human datasets, and assessed the extent of overlap between with each list. Gene enrichment set analysis was then used to determine whether the overlap was more than was expected by chance. We found that a majority of mouse model datasets we evaluated (namely, *Fmr1$^{-/y}$*, *Shank3$^{-/-}$*, *22q11.2$^{+/-}$*, *Cul3$^{-/-}$*) were enriched for the membrane protein alterations observed in 16p11.2$^{dup/+}$ mice. Human datasets for ASD and SZ were remarkably, also enriched, indicating a convergence in the human and mouse proteomic changes. We next evaluated if proteins from the epilepsy subnetwork were disrupted in these same human and mouse datasets. We found proteins from this network were also enriched in multiple mouse models (*Fmr1$^{-/y}$*, *22q11.2$^{+/-}$*, *Cul3$^{-/-}$*), and the level of enrichment in these models was consistently higher for the epilepsy network than for the global proteome disrupted in 16p11.2$^{dup/+}$ mice. In human post-mortem data, ASD-associated changes were more enriched in the epilepsy network than in the full 16p11.2$^{dup/+}$ membrane proteome (1.86 vs. 3.08 fold), as was the synaptosome dataset from SZ (1.99 vs. 3.34 fold). Together these data indicate the potential for shared proteomic pathophysiology across different genetic mouse models with diverse etiologies, and reveal that elements of the epilepsy network are disrupted in mouse models and human NPDs (Supplementary Fig. 2e, f).

In summary, we discovered a disrupted epilepsy module that is altered in NPDs and may regulate the activity of cortical networks in 16p11.2$^{dup/+}$ mice.

**Enhanced functional connectivity of primary cortical neuron networks from 16p11.2$^{dup/+}$ mice**
We hypothesized that the observed proteomic alterations could cause network alternations relevant to epilepsy and associated NPDs. In addition, the upregulation of synaptic gene sets suggests there may be alterations to network connectivity. Therefore, we used calcium imaging to simultaneously monitor the activity of hundreds of neurons from primary cortical cultures and establish the functional properties of neuronal networks from 16p11.2$^{dup/+}$ mice (Fig. 2a, b). We found that spontaneous neuronal network events in 16p11.2$^{dup/+}$ mice contained a higher fraction of co-active cells (Fig. 2c, d and Fig. 2e, *p* = 0.0239) but occurred at the same frequency as wild-type network events (Fig. 2f). We then treated neurons with bicuculline to evaluate whether altered network properties in 16p11.2$^{dup/+}$ mice were dependent on GABAergic signaling (Fig. 2g, h). We found that the larger proportion of co-active cells during network events remained after blockade of GABA$_A$ receptors, indicating that the higher co-activation rates were not caused by GABAergic neurotransmission. Together these data reveal that the 16p11.2$^{dup/+}$ causes an increase in the functional connectivity of cortical neuronal networks, due to a dysfunction of glutamatergic circuitry.

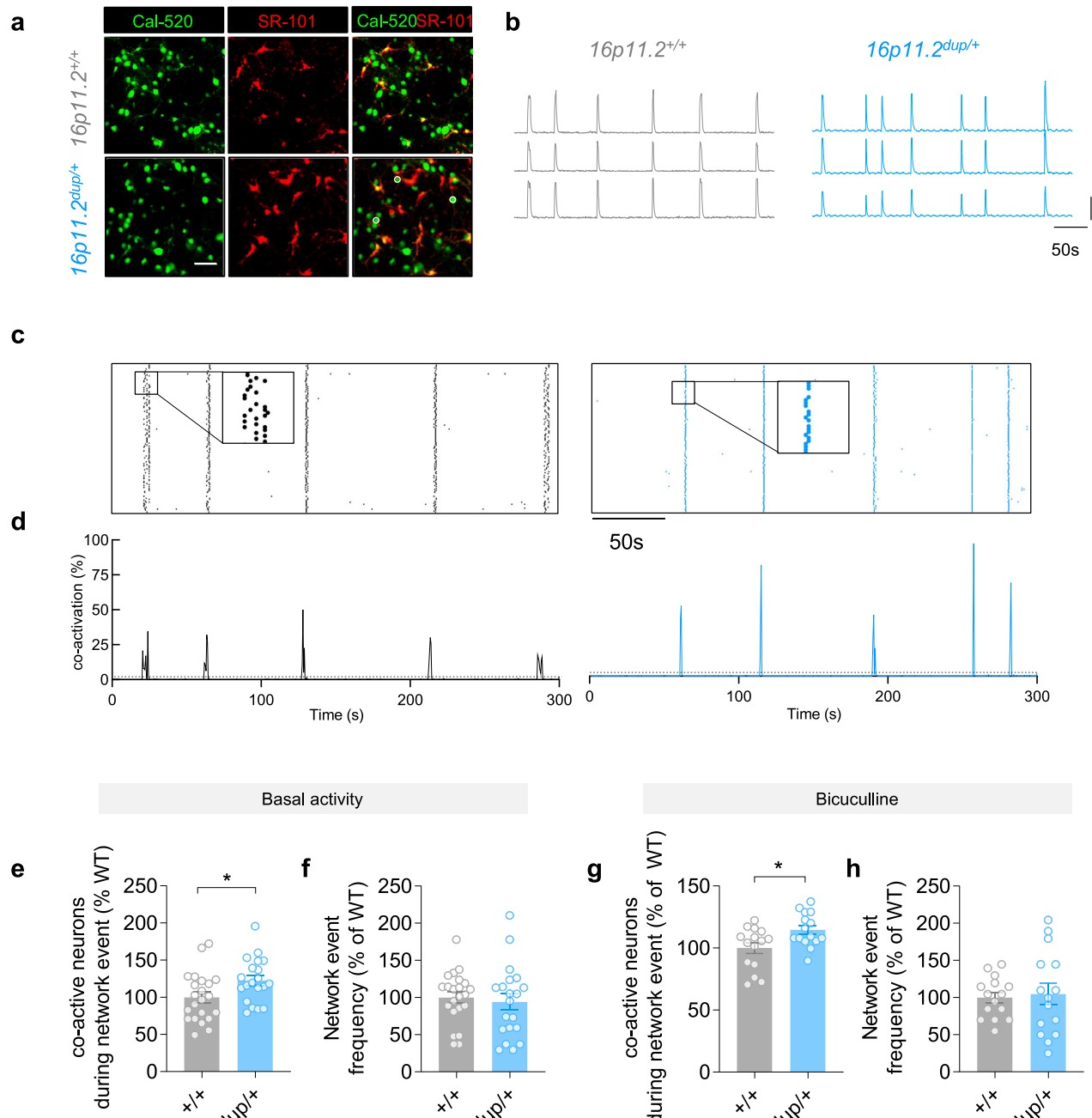

**Fig. 2 | Enhanced functional connectivity of primary cortical neurons from 16p11.2$^{dup/+}$ mice. a** Fluorescence microscopy images of cultured neuronal networks loaded with fluorometric calcium dye (Cal520, green) and astrocyte-specific dye Sulforhodamine 101 (SR101, red). Somatic ROIs used to measure calcium transients are shown. Scale bar, 50 μm. **b** Example calcium traces from individual neurons **c** Representative raster plots of network events in 16p11.2$^{+/+}$ (left) and 16p11.2$^{dup/+}$ (right) cultures. Each raster plot displays the activity of 150 neurons (y-axis) over 300 s (x-axis). **d** Quantification of neuronal co-activity (functional connectivity) during network events. **e**, **f** Properties of neuronal network activity under basal conditions (16p11.2$^{+/+}$, $n = 22$ FOV from 14 brains; 16p11.2$^{dup/+}$ = 20 FOV from 11 brains). **e** Increased number of co-active neurons during network events ($p = 0.0239$). **f** No change in network event frequency in 16p11.2$^{dup/+}$ neurons compared to 16p11.2$^{+/+}$ neurons ($p = 0.641$). **g–h** Properties of neuronal network activity when stimulated with 50 μM of bicuculline, a GABA$_A$ receptor inhibitor (16p11.2$^{+/+}$, $n = 15$ FOV from 5 brains; 16p11.2$^{dup/+}$ = 15 FOV from 6 brains). **g** Increased co-activation during network events ($p = 0.0136$) **h** No change in frequency of network events in 16p11.2$^{dup/+}$ neurons compared to 16p11.2$^{+/+}$ neurons. **i**. Data are shown as mean ± s.e.m. $^*p < 0.05$, $^{**}p < 0.01$, two-tailed $t$-test. Abbreviations: dup/+, 16p11.2$^{dup/+}$; +/+, 16p11.2$^{+/+}$; FOV, field of view.

## Cortical slices from 16p11.2$^{dup/+}$ mice exhibit hypersynchronous activity

Our studies in cultured neuron networks suggest that the dynamics of cortical circuits may be disrupted in 16p11.2$^{dup/+}$ mice. We therefore sought to evaluate dynamics of neurons within cortical microcircuits of acute brain slices, using two-photon calcium imaging (Fig. 3, Supplementary Fig. 4, Supplementary Movie 1). We recorded local calcium transients in the presence of bicuculline to stimulate cortical network activity. As expected, calcium events were fully dependent on action potentials as these were suppressed upon application of the voltage-gated sodium (Na$_v$) channel blocker, tetrodotoxin (Supplementary Fig. 4b). We found stark differences in calcium dynamics when evaluating individual wild-type and 16p11.2$^{dup/+}$ neurons (Fig. 3b, Supplementary Fig. 4c–e). Calcium peaks in 16p11.2$^{dup/+}$ neurons had a > 1.7

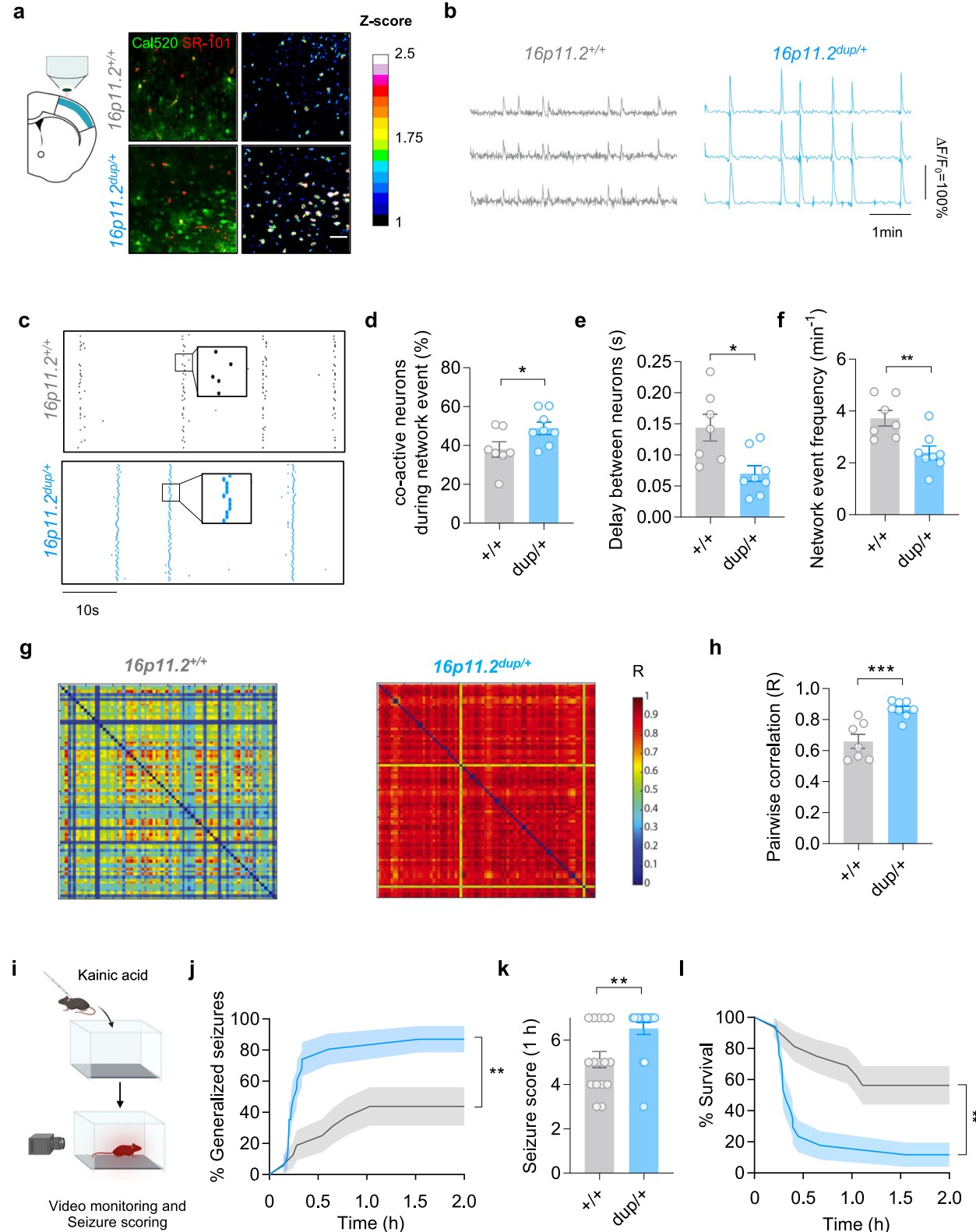

fold increase amplitude ($p = 0.01$), and an approximately 3 fold reduction in half-width ($p = 0.0052$) but no changes to the frequency of events was observed compared to wild-type neurons. We next assessed the circuit-level properties of 16p11.2$^{dup/+}$ mice (Fig. 3c–f). Here, we found that network events in 16p11.2$^{dup/+}$ mice were larger in magnitude, as they contained a higher number of simultaneously active cells (Fig. 3c, d) ($p = 0.0447$), mimicking the effects in neuronal culture experiments. To determine the propagation speed of calcium events, we measured the pairwise time delay between peaks of individual neurons, and observed a reduced duration between neuronal peaks of 16p11.2$^{dup/+}$ mice, indicating faster network activation (Fig. 3e) ($p = 0.0091$). We also found network events were lower in frequency compared to wild-type mice (Fig. 4f). Further analysis demonstrated that cortical neurons from 16p11.2$^{dup/+}$ mice displayed a high degree of

**Fig. 3 | Hypersynchronous activity in cortical slices carrying the 16p11.2 duplication. a** (Left) schematic showing the location of the primary somatosensory cortex (S1) imaged by two-photo calcium imaging. (Right) Representative images of a network event in 16p11.2$^{+/+}$ and 16p11.2$^{dup/+}$ somatosensory cortex. Images show labelling with a calcium-sensitive dye (Cal520, green), astrocytes specific dye (SR-101, red) and fluorescence intensity scaled by z-score. Scale bar, 50 μm. **b** Calcium transients of individual neurons. **c** Raster plots of network events in 16p11.2$^{+/+}$ and 16p11.2$^{dup/+}$ somatosensory cortex. Each raster plot displays the activity of 80 neurons (y-axis) over 50 s (x-axis) (**d**–**h**) Quantification of network events in S1 cortical slices (16p11.2$^{+/+}$, $n = 7$ slices from 4 mice; 16p11.2$^{dup/+}$, $n = 8$ slices from 4 mice). **d** Increased neuronal co-activation during network events ($p = 0.0447$) **e** Reduced time delay between neuronal pairs during network events ($p = 0.0091$). **f** Reduced number of synchronized network events in 16p11.2$^{dup/+}$ mice ($p = 0.0046$) **g** Representative matrices displaying degree of functional correlation between pairs of neurons (each matrix represents 80 × 80 neurons). The degree of correlation (R) is color-coded with a heatmap. **h** Increased pairwise correlation in S1 cortical circuits of 16p11.2$^{dup/+}$ mice, consistent with hypersynchronous activity ($p = 0.0007$). **i**–**l** Kainate (KA)-induced seizure experiment (16p11.2$^{+/+}$, $n = 16$ mice; 16p11.2$^{dup/+}$, $n = 17$ mice). **i** Schematic of KA-induced seizure experiment. Created with BioRender.com. **j** Kaplan–Meier curve showing the proportion of mice with generalized tonic-clonic seizures (GTCS). 16p11.2$^{dup/+}$ mice have a rapid onset and increased proportion of GTCS after kainate injection (Log-rank test, $p = 0.0031$, error bands represent s.e.m.) **k** Increased severity of seizures in 16p11.2$^{dup/+}$ mice at 1 h (Mann-Whitney test, $p = 0.0039$). **l** Kaplan–Meier curve reduced survival of 16p11.2$^{dup/+}$ mice after kainate induction (Log-rank test, $p = 0.0013$, error bands represent s.e.m.). Bar charts are displayed as mean ± s.e.m with $^*p < 0.05$, $^{**}p < 0.01$, $^{***}p < 0.001$, two-tailed $t$-test unless otherwise specified. Abbreviations: dup/+, 16p11.2$^{dup/+}$; +/+, 16p11.2$^{+/+}$.

correlated neuronal activity, consistent with enhanced functional connectivity and synchrony in the network ($p = 0.0007$) (Fig. 3g, h). Finally, we assessed whether these circuit phenotypes were distinct to the somatosensory cortex or present in other cortical structures. We performed a similar analysis in layer 2/3 of the primary visual cortex (Supplementary Fig. 4f–k). Here, we also found more correlated neuronal activity indicating that multiple cortical circuits may be excessively synchronous. Our data therefore reveals a core dysfunction of cortical circuitry in 16p11.2$^{dup/+}$ mice, with neuronal networks exhibiting hypersynchrony.

### 16p11.2$^{dup/+}$ mice are susceptible to induced-seizures
We reasoned that the abnormalities in circuit function may be associated with an increased propensity for seizure generation, because hypersynchronous circuits are closely linked to seizures[42]. However, we did not observe spontaneous convulsive seizures in 16p11.2$^{dup/+}$ mice during our study. Instead, we hypothesized that 16p11.2$^{dup/+}$ mice may have an altered susceptibility to induced seizures. We therefore injected 16p11.2$^{dup/+}$ mice with kainic acid, a well-established model to study seizure susceptibility (Fig. 3i). We found that both male and female 16p11.2$^{dup/+}$ mice were highly sensitive to seizure induction (Fig. 3j, $p = 0.0031$) (Supplementary Fig. 5). Overall, approximately 82% of 16p11.2$^{dup/+}$ mice exhibited generalized tonic-clonic seizures (GTCS) compared to approximately 44% of wild-type littermates ($p = 0.0324$). The 16p11.2$^{dup/+}$ mice had a more rapid onset of GTCS (Fig. 3j) and were associated with significantly higher seizure scores at 1 h post-injection (Fig. 3k, $p = 0.0039$). The increased severity of seizures resulted in only 12.5% of 16p11.2$^{dup/+}$ mice surviving the procedure compared to ~56 % of wild-type mice (Fig. 3l, $p = 0.0013$). These experiments demonstrate that the aberrant cortical circuit function in 16p11.2$^{dup/+}$ mice may promote susceptibility to seizures.

### Bioinformatic prediction of epilepsy subnetwork hub genes
Our data has shown that hypersynchrony is central network phenotype generated by neurons with the 16p11.2$^{dup/+}$, in cortical cultures and brain slices. These network properties may promote aberrant brain activity leading to NPDs. We hypothesized that the epilepsy-associated network would have a prominent role in regulating aberrant circuitry, given the link between epilepsy and excessive brain activity. Therefore, finding a regulator of this network could be an important avenue to modify disease phenotypes.

To identify a potential driver of the epilepsy subnetwork within the duplication region, we used a series of bioinformatics approaches, leveraging large human genetic and transcriptomic datasets[43] (Fig. 4a, b). We reasoned that network drivers would be functionally associated with proteins in the network. We, therefore, identified functionally associated co-expression networks of all genes within the 16p11.2 region (27 protein-coding) using developmental transcriptomic data from human cerebral cortex. We then compared the co-expression gene sets from all genes in the 16p11.2 region with the

proteomic data from the dysregulated proteins in the epilepsy subnetwork (Fig. 4a). Our analysis showed that the co-expression network associated with *SEZ6L2* (2.6 fold, $p = 2.2 \times 10^{-9}$) and *PRRT2* (2.52 fold, $p = 3.77 \times 10^{-8}$) were the most highly enriched for proteins in the epilepsy subnetwork, making these priority candidates (Fig. 4b). Because genes under evolutionary constraint are more likely to be associated with disease phenotypes, we assessed the constraint metrics (pLI) for each gene[44,45]. We found that *PRRT2* was among the top seven highest pLI scores (i.e. more constrained) whilst *SEZ6L2* had a comparatively low score (0.579 and 0.117 respectively) (Fig. 4b). We also considered Z-ratios from the membrane proteomic experiment in 16p11.2$^{dup/+}$ mice. Here, PRRT2 had a leading Z-ratio, indicating significant dysregulation in cortical membranes (Z-ratio=115) (Fig. 4b). Finally, we determined whether genes in the 16p11.2 region were associated with NPDs using data from exome sequencing studies of de novo variants and OMIM. We reasoned that genes associated with epilepsy or related NPDs would be more likely to regulate epilepsy-associated processes such as hypersynchrony, and could have a strong impact on pathophysiology in this model. The genetic analysis revealed that *PRRT2*, was a causative gene for an autosomal dominant form of epilepsy termed Benign Familial Infantile Seizures (MIM: 605751). Together this data suggests that PRRT2 may interact with the epilepsy subnetwork and impact downstream disease processes in 16p11.2$^{dup/+}$ mice.

### Interactome analysis reveals PRRT2 as a major hub in the epilepsy subnetwork
To validate our bioinformatic prediction, we aimed to determine whether PRRT2 can physically interact with proteins in the dysregulated epilepsy subnetwork. Although PRRT2 is known to interact with distinct SNARE proteins such as SNAP25 and ion channels[46–51], a comprehensive view of its interactome is lacking. We, therefore, undertook a global, discovery-based investigation of the PRRT2's protein:protein interactions in the mouse neocortex using large-scale immunoaffinity purifications combined with mass spectrometry (IAP-MS) (Fig. 4c). Our experiments identified a varied set of 208 PRRT2-associated proteins, most of which have never been identified before (Fig. 4d) (Supplementary Data 5). Importantly, we captured all known interactors or subunits thereof, including components of the SNARE complex (SYT1, VAMP2, STX1A), the Na$^+$/K$^+$ ATPase ATP1A3 and subunits of Ca$_v$ channels (CACNB1-2, CACNB4), AMPA receptors (GluA2, TARP-ϒ8) and Na$_v$ channels (SCN1B, SCN2B), validating our approach. A subset of interactions with SNAREs and Na$_v$ channel components were confirmed by overexpression in heterologous cells followed by immunoprecipitation (IP) and Western blotting, or by IP in cortical membrane fractions (Supplementary Fig. 6). To independently assess the quality of the interactome, we also intersected the full list of PRRT2-associated proteins identified by MS with the top genes co-expressed with *PRRT2* in the developing human neocortex, two independent datasets that should converge onto proteins functionally related to PRRT2.

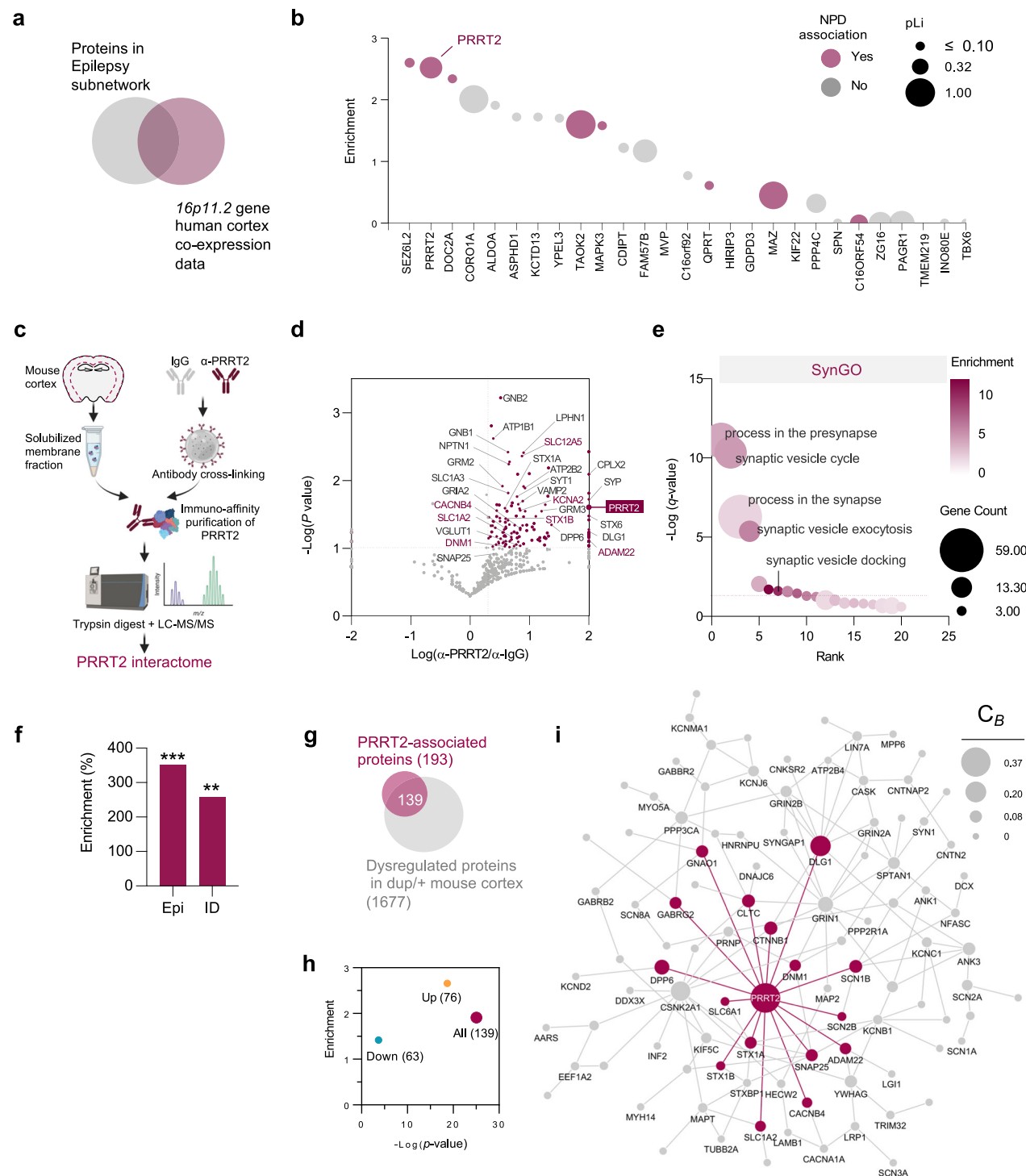

Remarkably, these datasets shared a 32.3% overlap (1.82 fold enrichment, $p = 3.79 \times 10^{-7}$), strengthening the confidence in our interactome dataset (Supplementary Fig. 7a).

We found that the interactome dataset was highly enriched for biological processes involving synaptic signaling and presynaptic processes, mirroring the pathways disrupted in the 16p11.2$^{dup/+}$ mouse neocortex (Supplementary Fig. 7b, Fig. 4e) (Supplementary Data 6 and 7). Gene set analysis revealed a major enrichment of proteins encoded by OMIM epilepsy genes (+252%, $p = 3.14 \times 10^{-5}$) and ID genes (+157%, $p = 5.00 \times 10^{-3}$) suggesting it operates within a biological network with other epilepsy-associated gene products (Fig. 4f) (Supplementary Fig. 7c).

The identification of the PRRT2 interactome in the neocortex allowed us to determine whether proteins dysregulated in the 16p11.2$^{dup/+}$ proteome could interact with, and therefore could be influenced by, PRRT2 on a global scale. We therefore compared the PRRT2 interactome with the altered cortical proteome of 16p11.2$^{dup/+}$ mice. We conducted both sets of experiments within the same brain region, and at the same developmental time point, to ensure they would be amenable to direct comparison. Remarkably, we show that PRRT2-interacting proteins were systematically disrupted in the membrane proteome of 16p11.2$^{dup/+}$ mice (72% of interacting proteins) (Fig. 4g), and conversely 8.3% of the disrupted membrane proteome were part of PRRT2-complexes. This overlap represents an enrichment

**Fig. 4 | PRRT2 is a synaptic and epilepsy network hub. a** Human brain co-expression networks for each gene were overlapped with proteins in the epilepsy subnetwork (Fig. 1e) to identify potential network hubs. **b** Bubble plot showing enrichment results from gene set analysis of (**a**) on the y-axis. Bubble size is scaled by the constraint metric for that gene (pLI) according to the gnomAD database [109] and colored by disease association. Higher pLI scores indicate more intolerance to mutation. Genetic associations are defined as genes with at least one de novo single nucleotide variant (dnSNVs) in a coding region from EE, ID, ASD or SZ probands, or genes associated with a brain-related disorder in the OMIM database. **c** Schematic of immunoaffinity purification combined with mass spectrometry (IAP-MS) used to identify the PRRT2 interactome. Created with BioRender.com. **d** Volcano plot of proteins identified in the PRRT2 IAP-MS experiment (*n* = 3 independent experiments). All proteins with spectral counts enriched >2 fold compared to IgG and *p*-value <0.1 (one-tailed *t*-test), were considered part of the interactome. A fold change of $\log_{10}(2)$ indicates that the protein was not identified in the control IgG IAP. Colored protein names indicate epilepsy-associated genes **e** SynGO analysis of the PRRT2 interactome (one-sided Fisher's Exact test with FDR-correction) **f** Epilepsy and intellectual disability associated proteins (from the OMIM database) are enriched in the PRRT2 interactome dataset (Epi, *p* = $3.14 \times 10^{-5}$; ID, *p* = $5.01 \times 10^{-3}$). Data shows percentage enrichment and *p*-values from a two-sided hypergeometric test. **g** 72% of PRRT2-interacting proteins are dysregulated in the neocortex of the 16p11.2$^{dup/+}$ mouse model **h** Enrichment values from a hypergeometric test of (**g**). **i** Novel interactions identified in the PRRT2 interactome were mapped onto the proteomic subnetwork identified in Fig. 1f. Grey edges represent retrieved protein interactions, whereas PRRT2 interactions identified in the IAP-MS experiment are colored as purple edges/nodes. Nodes are scaled by betweenness centrality ($C_B$) score to highlight nodes with strong influence on the protein interaction network. Epi epilepsy, ID intellectual disability, SZ schizophrenia, ASD autism spectrum disorder, DD developmental disorder, $P_{corr}$, Benjamini-Hochberg corrected *p*-value.

of 1.9-fold (*p* = $8.56 \times 10^{-26}$) over what would be expected from the background membrane proteome, and strongly suggests that PRRT2 regulates protein networks within the cortical proteome of 16p11.2$^{dup/+}$ mice (Fig. 4h). Further, we examined whether the PRRT2 interactome could directly interact with proteins in the epilepsy subnetwork identified in 16p11.2$^{dup/+}$ mice. We compiled a list of binary interactions identified in the IAP experiment and merged the PRRT2 interactome with the epilepsy subnetwork from Fig. 1, generating a new network (Fig. 4i). By performing network analysis, we found that PRRT2 had the highest degree of connectivity (*D* = 16) and betweenness centrality ($C_B$ = 0.37) in this network, making it the most topologically important node, and central hub in the epilepsy subnetwork of 16p11.2$^{dup/+}$ mice (Fig. 4i). Thus, our data strongly implicates hub protein PRRT2 as a regulator of the epilepsy subnetwork disrupted in 16p11.2$^{dup/+}$ mice, and suggests it may impact core disease phenotypes.

### Increased PRRT2 dosage and neurological disorders

Genetic studies have implicated heterozygous loss-of-function mutations in *PRRT2* with infantile epilepsy and paroxysmal movement disorders (MIM: 605751and 128200), however it is not known what the impact of a *PRRT2* gain-of-function may be. To determine if increased *PRRT2* copy number alone may have clinical relevance, we also considered whether *PRRT2* duplications existed in the absence of a full 16p11.2$^{dup/+}$ in humans, and if these would be sufficient for a neurological phenotype. We, therefore, searched the ClinVar database and found four patients with duplications encompassing the entire *PRRT2* coding region (Supplementary Fig. 7d, e). Interestingly, two of these patients had paroxysmal dyskinesia and two patients exhibited seizures, suggesting that increased *PRRT2* gene dosage alone may be able to cause neurological symptoms in humans.

### Prrt2 correction restores cortical network synchrony and glutamate release in 16p11.2$^{dup/+}$ mice

To experimentally test the involvement of PRRT2 in disease phenotypes, we employed mouse genetics to selectively correct *Prrt2* expression in the 16p11.2$^{dup/+}$ mouse model. We crossed 16p11.2$^{dup/+}$ mice with mice harboring heterozygous *Prrt2* deletions (*Prrt2*$^{+/-}$ or Het), producing offspring with the duplication but corrected for *Prrt2* copy number (16p11.2$^{corr}$ mice) (Supplementary Fig. 8a). This strategy constitutively adjusted PRRT2 expression in 16p11.2$^{dup/+}$ mice to wild-type levels, leaving other duplicated proteins increased (Supplementary Fig. 8b, c). We subsequently used mice from this cross to investigate the function of cortical networks in 16p11.2$^{corr}$ mice.

The data from the PRRT2 interactome suggested a close relationship between PRRT2, presynaptic function and biological pathways relevant to epilepsy. We, therefore, used a genetically encoded glutamate sensor exclusively expressed in neurons (Syn-iGluSnFR) to assess the function of cortical networks in 16p11.2$^{dup/+}$ mice corrected for *Prrt2* expression[52] (Supplementary Movie 2).

Using a glutamate sensor instead of Ca$^{2+}$ imaging would allow us to directly visualize synaptic release at the network level, as opposed to monitoring Ca$^{2+}$ dynamics in the soma. Cortical network activity acquired through iGluSnFR imaging revealed a rich repertoire of spontaneous events, comprising both high (>0.2 ΔF/F$_0$) and low (<0.2 ΔF/F$_0$) synchrony events (Fig. 5a, b). Importantly, iGluSnFR fluorescence was action potential-dependent, and sensitive to increasing glutamate concentrations (Supplementary Fig. 8d, e). Interestingly, high synchrony events were strongly reminiscent of large network events captured by Ca$^{2+}$ imaging, in both their magnitude and event rate (Fig. 5b, Fig. 2c, d). Whilst all events, or low synchrony events, were unchanged between genotypes, high synchrony events had a larger amplitude in 16p11.2$^{dup/+}$ mice compared to wild type (Fig. 5c–g) (Supplementary Movie 2). High synchrony events were also reversed upon correcting *Prrt2* copy number, demonstrating their sensitivity to PRRT2 dosage (Fig. 5f). We next used KCl-induced depolarization to determine whether evoked glutamate release was altered (Fig. 5h). We found an increased in the peak amplitude of KCl-evoked glutamate release in cortical networks from 16p11.2$^{dup/+}$ mice, and this phenotype was also reversed in corrected mice (Fig. 5i, j). The peak area and half-life of the decay were not different between genotypes (Fig. 5k, l). These data establish PRRT2 as a regulator of cortical network synchrony and glutamatergic function in 16p11.2$^{dup/+}$ mice.

### Correcting Prrt2 gene dosage rescues neuropsychiatric phenotypes in 16p11.2$^{dup/+}$ mice

We next assessed the effect of *Prrt2* correction on behavioral phenotypes, established both here and in previous work[29,30]. Behavioral analysis revealed that 16p11.2$^{dup/+}$ mice have decreased locomotor activity in the open field, reproducing previous findings (Fig. 6a–c). However, *Prrt2*-corrected mice traveled a similar distance to 16p11.2$^{dup/+}$ mice, indicating that *Prrt2* dosage has no effect on locomotor activity (Fig. 6a–c). We found no evidence of anxiety phenotypes, as assessed by time spent in the center of the open field area, in agreement with previous work (Fig. 6c)[30].

To assess the effects of *Prrt2* copy number on seizure phenotypes, we subjected mice to the same chemo-convulsant seizure paradigm used previously. Here, we found that *Prrt2*$^{+/-}$ mice had a heightened sensitivity to seizures, in support of previously published data using similar assays (Fig. 6d–f)[53]. The 16p11.2$^{dup/+}$ mice were similarly susceptible to seizures replicating our previous findings (Fig. 6d). Remarkably, correction of PRRT2 expression reversed seizure phenotypes in 16p11.2$^{dup/+}$ mice (*p* = 0.0231, Fig. 6d) resulting in a GTCS onset curve that was statistically indistinguishable from wild-type mice (Fig. 6d). This effect was due to a delay in the onset of GTCS, allied with a reduced proportion of mice that manifested GTCS. The 16p11.2$^{corr}$ mice had an approximately 40% delay in GTCS onset compared to 16p11.2$^{dup/+}$ mice (*p* = 0.0227, Supplementary Fig. 8f). Additionally, the

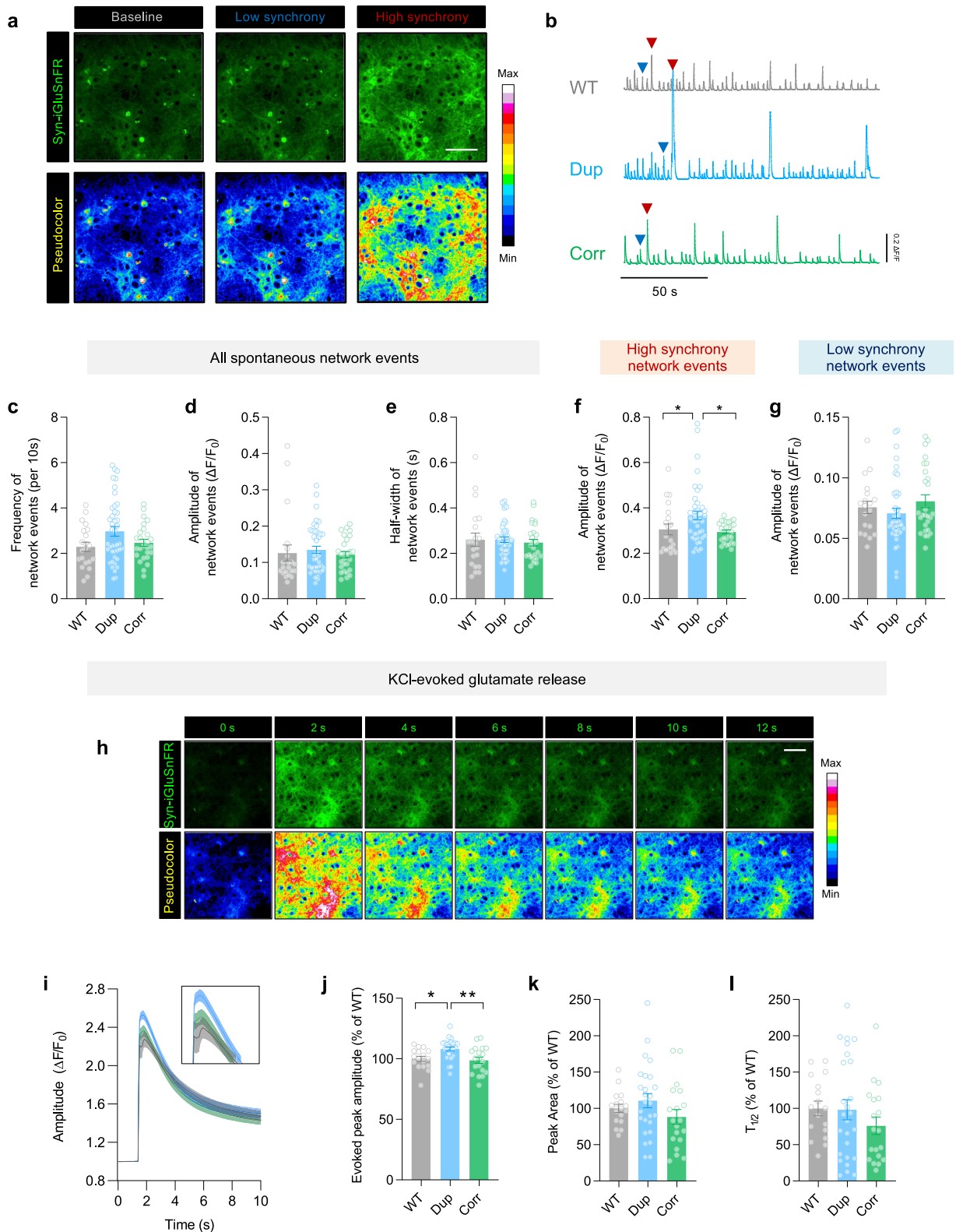

severity of seizures in 16p11.2$^{corr}$ mice was rescued to wild-type levels at 1 h post-injection, and resulted in higher survival rates compared to mice with an intact 16p11.2$^{dup/+}$ (-81% compared to -62%) (Fig. 6e, f). Thus, genomic correction of *Prrt2* resulted in a delayed onset and reduced severity of seizures in 16p11.2$^{dup/+}$ mice. These results indicate that both exaggerated and insufficient *Prrt2* expression converge seizure susceptibility.

Finally, we investigated the effects of *Prrt2* on the social deficits reported in 16p11.2$^{dup/+}$ mice, a core behavioral endophenotype associated with ASD and SZ. Mice were subjected to the three-chamber social interaction test, to assess the sociability of each genotype[54] (Fig. 6g). During the habituation phase, mice did not show a preference for either side of the chamber (Supplementary Fig. 8g). In the sociability phase, all mice had a preference for the social cue (Fig. 6h, i).

**Fig. 5 | Aberrant network synchrony and excess glutamate release in primary cortical neurons from 16p11.2$^{dup/+}$ mice are restored upon correcting *Prrt2* copy number. a** Epifluorescence images of the glutamate sensor Syn-iGluSnFr infected in cortical neurons (Scale bar 100 µm). Examples of high (>0.2 ΔF/F$_0$) and low synchrony (<0.2 ΔF/F$_0$) events are illustrated **b** Individual traces showing network-level changes in iGluSnFr fluorescence. High and low synchrony events are indicated with red and blue arrows respectively. **c–g** Spontaneous glutamate imaging of cortical cultures (WT = 20 FOV from 6 brains; Dup = 42 FOV from 11 brains; Corr = 27 FOV from 10 brains; Kruskal–Wallis test with Dunn's post-hoc test). **c–e** Cortical networks from 16p11.2$^{dup/+}$ mice do not show changes in network event frequency, amplitude or half-width. **f** The amplitude of high synchrony events is increased in 16p11.2$^{dup/+}$ mice ($p = 0.0378$), and is restored to WT levels in corrected mice ($p = 0.0382$). **g** The amplitude of low synchrony events remain unaltered between genotypes. **h** Fluorescence changes after evoked glutamate release using KCl (30 nM). **i–l** Evoked glutamate release in cortical cultures (WT = 17 FOV from 8 brains; Dup = 26 FOV from 12 brains; Corr = 19 FOV from 10 brains; One-way ANOVA with Holm-Šídák's post-hoc test) **(i, j)** Higher amplitude of evoked events in 16p11.2$^{dup/+}$ mice ($p = 0.0185$) are rescued in corrected mice ($p = 0.0063$). **k-l** The *p*eak area and half-life (T$_{1/2}$) of the evoked glutamate release are unaltered between genotypes. Data are displayed as mean ± s.e.m with $^*p < 0.05$; $^{**}p < 0.05$. Abbreviations: WT, 16p11.2$^{+/+}$; Dup, 16p11.2$^{dup/+}$; Corr, 16p11.2$^{corr}$.

However, 16p11.2$^{dup/+}$ mice had a sustained deficit in sociability when compared to wild-type mice (DI (%) = 34.67 vs 60.11, respectively) (Fig. 6j). *Prrt2*$^{+/-}$ mice conversely, had no social deficit. Strikingly, behavioral analysis in 16p11.2$^{Corr}$ mice revealed a complete restoration of social behavior (DI (%) = 61.59) (Fig. 6j).

Together, these results indicate that correction of PRRT2 dosage can rescue distinct circuit and behavioral phenotypes relevant to neuropsychiatric disorders in 16p11.2$^{dup/+}$ mice.

## Discussion

NPDs have overlapping symptoms and often co-occur with epilepsy, suggesting the existence of convergent biology, but the underlying molecular and circuit mechanisms remain unknown. A powerful entry point to understand shared risk mechanisms is to study highly penetrant genetic variants, that associate with NPDs and epilepsy. In this work, we use a mouse model of the 16p11.2 duplication to reveal key protein networks and circuit phenotypes which may be involved in mediating risk for a spectrum of NPDs. Importantly, we identify *Prrt2* as a single gene capable of reversing important disease-associated phenotypes such as seizure susceptibility and social dysfunction.

### Membrane proteomics reveals an epilepsy network disrupted in NPDs

A major challenge in understanding disease processes is to resolve the mechanisms between genotype and phenotype. Although it has been well established that CNVs, such as the 16p11.2 duplication, exert their effects via changes to transcript levels of affected genes, how these gene-dosage effects lead to more global alterations, especially in proteomic networks, is poorly understood. Transcriptomic approaches are a key step in understanding disease mechanisms. However, proteins are the fundamental components of neuronal circuits, and mRNA is a poor predictor of protein abundance in dynamic systems[55]. In addition, cellular compartments have discrete levels of mRNA and protein to serve specialized functions such as synaptic transmission[56]. To assess the level of overlap between RNA and protein, we compared our membrane proteome data to previously published RNA-seq datasets in the same mouse model. We found little to no correlation between changes in mRNA and protein, exemplifying the importance of performing proteomic analysis in addition to transcriptomics, to fully capture the disease process. The lack of correlation between RNA and proteins suggests that although transcriptional effects may explain some aspects of disease, they cannot explain how membrane proteins are altered in this model. It is therefore likely that post-transcriptional effects regulating protein abundance and localization are involved. Given that PRRT2 has been show to regulate surface trafficking of membrane proteins[48,49], and that it interacts with an abundance of proteins that are dysregulated, we would propose that PRRT2-dependent membrane trafficking could be an important player in remodeling the membrane proteome of 16p11.2$^{dup/+}$ mice.

Our discovery-based proteomic screen uncovered that disrupted proteins are associated with a wide range of disorders including ASD and SZ. However, we found that protein networks associated with epilepsy were particularly enriched amongst dysregulated proteins.

These data suggest a molecular hierarchy may exist whereby epilepsy networks are preferentially affected within membranes of 16p11.2$^{dup/+}$ mice. Importantly, we found that proteins in the dysregulated epilepsy network were also disrupted in NPD mouse models with distinct genetic etiologies, and in human brain tissue from individuals with ASD and SZ. These data may be revealing of a molecular signature associated with NPDs. Thus, we show proteomic alterations in epilepsy networks may be key to the pathophysiology in 16p11.2$^{dup/+}$ mice.

### Circuit phenotypes in 16p11.2 duplication syndrome mice

Synchronous activity in neuronal networks occurs when populations of cells fire in unison, creating correlated activity. The synchronization of neurons in local and long-range circuits is an essential mechanism for information processing in the brain[57,58]. Here, we find an abnormally high number of co-active neurons during large network events, indicating enhanced functional connectivity and synchrony. We demonstrate circuit dysfunction 16p11.2$^{dup/+}$ mice using Ca$^{2+}$ and glutamate imaging, two complementary techniques, converging on alterations to large synchronized network events. Importantly, we show that circuit hypersynchrony occurs even after blockade of GABAergic inputs, demonstrating that these properties are independent of inhibitory synapses. However, the deficits in inhibitory synapses previously reported may exacerbate the abnormal glutamatergic circuitry, leading to a further circuit dysfunction in vivo[30]. The increased neuronal co-activation phenotype is observed in cultured neuron networks, somatosensory cortex and visual cortex, suggesting that heightened synchrony could be a fundamental property of cortical circuits in 16p11.2$^{dup/+}$ mice.

### Implications of circuit phenotypes for epilepsy

Epilepsy is defined as the presence of recurrent, unprovoked seizures, which can be captured on electroencephalograms (EEGs). Epileptiform activity recorded on EEGs are thought to arise from the uncontrolled synchronized firing of thousands of neurons. Despite elevated synchrony being associated with seizures[42], we did not observe any overt convulsive seizures in 16p11.2$^{dup/+}$ mice. Instead, we show that 16p11.2$^{dup/+}$ mice are susceptible to chemically-induced seizures, which suggests the presence of a sub-threshold circuit phenotype. We propose that the elevated synchrony of cortical circuits increases the risk for seizure generation, which could cause epilepsy in specific genetic and/or environmental contexts. Evidence from brain imaging studies supports the view that network hypersynchrony is an epilepsy endophenotype and may facilitate epileptogenesis[59]. It is also noteworthy that not all human 16p11.2$^{dup/+}$ carriers have seizures. However, seizures are one of the most prevalent symptoms in 16p11.2$^{dup/+}$ carriers affecting up to 40% of individuals[26]. Interestingly, the 16p11.2$^{dup/+}$ confers high risk Rolandic epilepsy, which is one of the most common forms of childhood epilepsy, affecting sensorimotor areas[20]. Thus, increased synchrony in the somatosensory cortex area may increase risk for Rolandic seizures. Nonetheless, 16p11.2$^{dup/+}$ carriers are reported to have a diverse seizure types including, focal dyscognitive, generalized tonic-clonic, febrile and infantile seizures, indicating a seemingly general susceptibility to seizures[25,26].

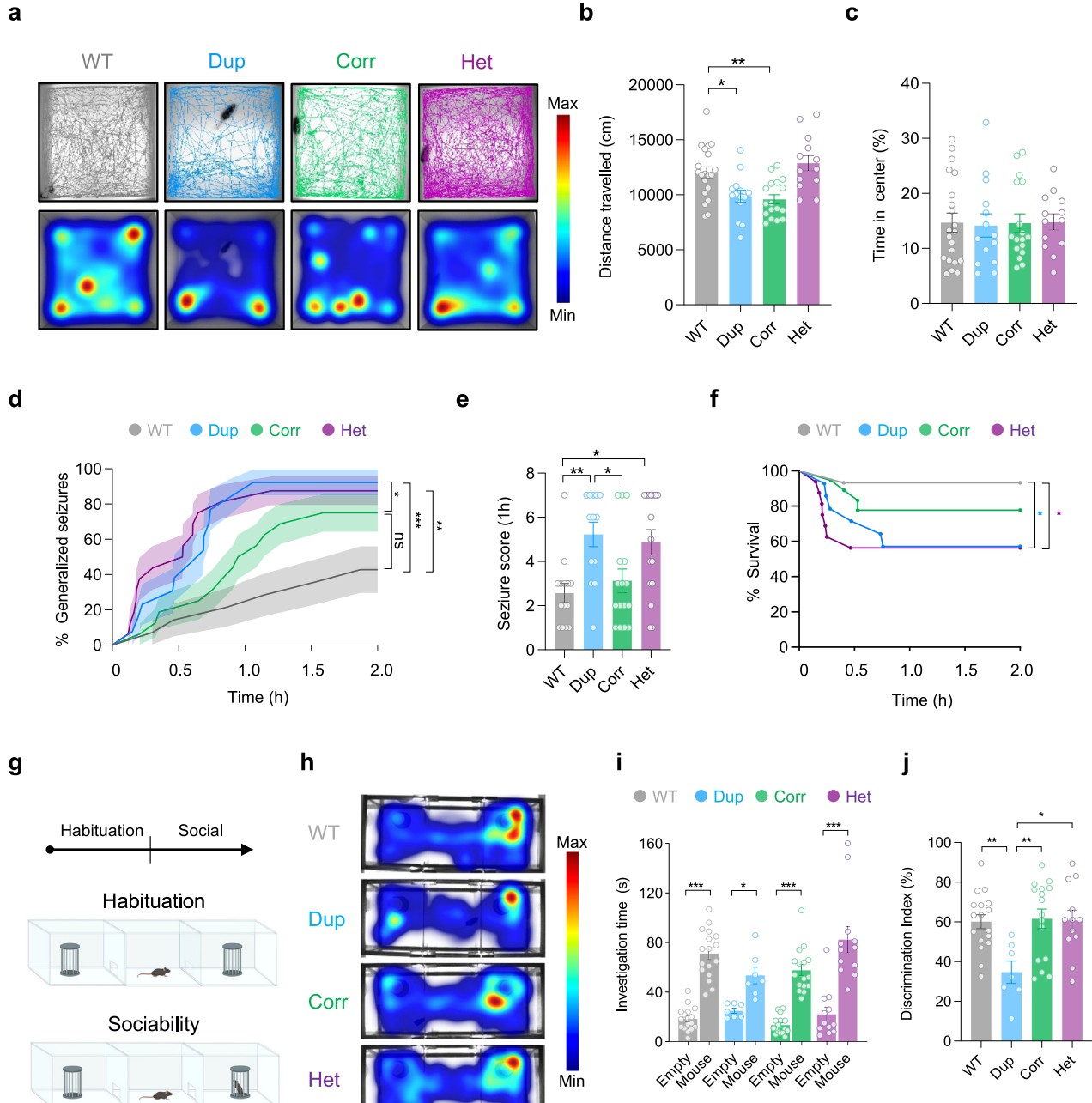

**Fig. 6 | Genetic correction of *Prrt2* rescues neuropsychiatric phenotypes in 16p11.2^dup/+ mice. a** Traces and heatmaps of locomotor activity of mice during 30 min recordings. **a–c** Dup and Corr mice have reduced locomotor activity compared to WT ($p = 0.0323$ and $0.0073$, respectively), and do not appear to have anxiety-like phenotypes (WT, $n = 20$ mice; Dup, $n = 14$ mice; Corr, $n = 17$ mice; Het, $n = 13$ mice) **d, f** Correction of PRRT2 gene dosage reverses seizure phenotypes of 16p11.2^dup/+ mice (WT, $n = 14$ mice; Dup, $n = 13$ mice; Corr, $n = 16$ mice; Het, $n = 16$ mice). **d** Reversal of seizure phenotypes is manifested by a delay in the onset of generalized tonic-clonic seizures (GTCS), and a reduced percentage of mice with GTCS (WT vs. Dup, $p = 0.0006$; WT vs. Het, $p = 0.0012$; Dup vs. Corr, $p = 0.0223$, Log-rank test, error bands represent s.e.m.). **e** Seizure severity in 16p11.2^corr mice is comparable to WT mice at 1 h post-injection (WT vs. Dup, $p = 0.0074$; WT vs. Het, $p = 0.017$; Dup vs. Corr, $p = 0.039$). **f** 16p11.2^corr mice have improved survival after

induced seizures due to reduced seizure severity (WT vs. Dup, $p = 0.025$; WT vs. Het, $p = 0.017$) Log-rank test, for clarity error bands are not shown). **g–j** Social behavior (WT, $n = 17$ mice; Dup, $n = 7$ mice; Corr, $n = 16$ mice; Het, $n = 12$ mice). **g** Schematic of apparatus used for three-chamber social interaction test. Created with BioRender.com. **h** Heatmaps of activity during social behavior. **i** Amount of time each test mouse spent investigating the novel mouse or empty cup (WT, $p < 0.0001$; Dup, $p < 0.0212$; Corr, $p < 0.0001$; Het, $p < 0.0001$). **j** Dup mice have social deficits compared to WT and PRRT2-correction restores social behavior (WT vs. Dup, $p = 0.0086$; Dup vs. Corr, $p = 0.0054$; Dup vs. Het, $p = 0.0114$). Bar charts are displayed as mean ± s.e.m with $^*p < 0.05$, $^{**}p < 0.01$, $^{***}p < 0.001$ (Two-way ANOVA with Tukey's post-hoc test). Abbreviations: WT, 16p11.2^+/+; Dup, 16p11.2^dup/+; Corr, 16p11.2^corr; Het, *Prrt2*^+/-.

## Implications of circuit phenotypes for other NPDs

Alterations in neuronal synchrony have been associated with a wide variety of NPDs including autism and schizophrenia[60]. The disruption of correlated activity in neural networks located in discrete brain areas

could give rise to a range of behavioral symptoms exhibited by individuals with NPDs, by disrupting the sparse coding patterns, increasing noise or impairing the processing of salient signals[61-64]. For example, excessive synchrony in local somatosensory and visual cortices could

provoke enhanced sensory sensitivity exhibited by 16p11.2 duplication carriers[65]. Alternatively, aberrant synchrony may contribute to 'noisy networks' which impair information processing and undermine higher-order cortical processing[66–68]. Interestingly, analysis of resting-state EEGs has shown that 16p11.2 duplication carriers have elevated synchrony in the theta band, an oscillation linked to cognition, learning and memory[69]. In addition, $Ca^{2+}$ imaging studies in syndromic ASD and SZ mouse models have found increased neuronal co-activity in cortical circuits, suggesting that hypersynchronous activity may be a convergent mechanism in NPDs co-morbid with epilepsy[70–73]. Overly synchronized cortical circuits therefore represent an attractive mechanism by which multiple neuropsychiatric phenotypes could be affected including aspects of cognition, social functioning and seizures. Here, we have shown that reversing circuit phenotypes in the 16p11.2$^{dup/+}$ mice is also associated with a rescue of social deficits, a non-epileptic phenotype. However, the relevance of the circuit phenotype to the wide variety of symptoms exhibited in human carriers, as well as the exact brain regions and circuits involved, remains to be fully determined.

### PRRT2 and disease mechanisms in 16p11.2 duplication mice

The 16p11.2 region contains 30 genes, many of which are expressed in the brain, making it likely that one or more genes contribute to disease phenotypes. Although many candidate genes have been but forward, empirical evidence of their role at the behavioral level, and in the context of the disease model, remains scarce. Here, we show that PRRT2 has a critical role both at the circuit and behavioral level, modulating disease-relevant phenotypes in 16p11.2$^{dup/+}$ mice. Our work suggests that increased *Prrt2* gene dosage causes an upregulation of PRRT2 at the membrane, which dysregulates epilepsy-associated protein networks, causes elevated circuit synchrony, increased glutamate release, which then impacts both seizure susceptibility and sociability.

Intriguingly, loss-of-function mutations in *PRRT2* (as opposed to gain-of-function in the duplication) cause benign familial infantile seizures (OMIM: 605751), one of the most common infantile seizure disorders. Our work is therefore unexpected given that reducing PRRT2 levels is associated with seizures and neurological symptoms, but reducing PRRT2 in the 16p11.2$^{dup/+}$ mouse model has beneficial effects on seizure susceptibility and behavioral phenotypes. This suggests that both excessive or insufficient levels of *PRRT2* may be detrimental and cause neuropsychiatric symptoms, with increased *Prrt2* dosage being associated with impaired social behavior, which has not been implicated by loss-of-function studies. Despite this, care must be taken to draw parallels between human and mouse phenotypes. The 16p11.2$^{dup/+}$ mouse model has strong construct validity because the same set of genes are duplicated in mice compared to human conditions. However, it is not possible to discern whether behavioral deficits exhibited in mice (face validity) are a true reflection of social impairments observed in NPDs. Sociability alterations in 16p11.2$^{dup/+}$ mice could represent phenocopies of human symptoms that are not mechanistically linked[74]. Nevertheless, social deficits appear to be one of the most reproducible behavioral phenotypes in 16p11.2$^{dup/+}$ mice and are rescued by correcting PRRT2 expression.

Previous work has shown that PRRT2 is a synaptic protein enriched in presynaptic terminals of glutamatergic neurons[47,75]. This subcellular expression pattern is highly relevant to our work, given that gene sets associated with excitatory neurons and presynaptic function were preferentially disrupted in the 16p11.2$^{dup/+}$ membrane proteome. Using IAP-MS in the mouse neocortex, we showed that PRRT2 interacts with a broad range of proteins, enriched in presynaptic function and products of ID and epilepsy genes. This data is consistent with previous reports showing that PRRT2 associates with members of the SNARE complex, and regulates $Ca^{2+}$-dependent fusion of synaptic vesicles[47]. However, our proteomic data goes beyond previous studies,

showing over 200 proteins linked to PRRT2 function which we hope will be valuable information to study this important disease gene in its own right. Interestingly, PRRT2 expression begins perinatally in mice, suggesting that the rescue mechanisms we observe are largely post-natal and occur after the major milestones of cortical development[47]. These data demonstrate that postnatal therapies may be effective in 16p11.2 duplication carriers and that targeting single genes may also be applicable strategy in CNV disorders[76].

Collectively, our work illustrates that proteomics can aid the identification of disease hubs capable of phenotype reversal in complex CNVs, and underscores molecular and circuit phenotypes that may underlie NPDs co-morbid with epilepsy.

## Methods

### Mouse models

All animal procedures were performed with the approval of the Institutional Animal Care and Use Committee (IACUC) at Northwestern University (IS00006149 and IS00015320). The 16p11.2$^{dup/+}$ mouse model carrying a duplication of mouse chromosome 7qF4 (the syntenic region of human chromosome 16p11.2) was generated by Dr. Alea Mills (Cold Spring Harbor Laboratory)[28]. Mice were backcrossed >5 generations to a C57BL/6 J background by Dr. Nicolas Katsanis and kindly gifted to our laboratory. Male 16p11.2$^{dup/+}$ mice were then crossed for at least 2 generations on a C57BL/6 N background (Charles River Laboratories) prior to experiments. We did SILAM labeling as previously described[77]. Briefly, female mice (F0) were fed with a $^{15}$N-rich Spirulina-based food for ten weeks starting at P21 through breeding and weaning. In the brains from the "heavy" litters (F1), at least 95% of proteins were $^{15}$N enriched. The 'Heavy' F1 mice were fed with a normal ($^{14}$N) diet for ten weeks. *Prrt2*$^{+/-}$ on a C57BL/6 J background were generated from *Prrt2*$^{fl/fl}$ and kindly donated by Mark LeDoux[78]. For *Prrt2* correction experiments, 16p11.2$^{dup/+}$ mice were crossed with *Prrt2*$^{+/-}$ mice. Mice were housed according to gender on a 14 h on/10 h off light/dark cycle with temperature 70–74 °F and humidity 30–70%. Up to five animals were housed per cage, with mixed-genotype littermates. All experiments were performed on mice of both sexes at 6–10 weeks of age unless otherwise stated. All experiments were performed on littermate controls.

### Western blotting

or western blotting of proteins encoded in the 16p11.2 region, cortical lysates were prepared by solubilizing in cold RIPA buffer (50 mM Tris pH 7.4, 150 mM NaCl, 5 mM EDTA, 1% Triton X-100, 0.5% sodium deoxycholate, 0.1% SDS, + protease inhibitors added fresh) for 1 h prior to SDS-PAGE and western blotting with anti-SEZ6L2 (Abcam, ab197058, 1:500), anti-PRRT2 (Sigma, HPA014447, 1:2000), anti-ERK1 (Santa Cruz, sc94, 1:5000), anti-TAOK2 (Santa Cruz, sc-47447, 1:500), anti-FLAG (Sigma, F1804, 1:1000), anti-T7 (Millipore, AB3790, 1:1000), anti-Myc (Santa Cruz, sc-789, 1:1000) antibodies

### Fractionation for membrane proteomics

Cortical hemispheres from 6-week old male 16p11.2$^{dup/+}$ mice ($n = 5$), their 16p11.2$^{+/+}$ male littermates ($n = 5$), and $^{15}$N-labeled mice were dissected and flash frozen in liquid nitrogen. We next prepared P2 membrane fractions (or crude synaptosomes) from the frozen cortical tissue. All cortices were homogenized in cold sucrose buffer (20 mM HEPES pH 7.4, 320 mM sucrose, 5 mM EDTA) supplemented with protease inhibitor cocktail (Roche) and a BCA was perform to assess the protein concentration of each sample. Following quantification, $^{15}$N-labeled homogenates were added in a 1:1 ratio individually to each 16p11.2$^{dup/+}$ and 16p11.2$^{+/+}$ mouse homogenate. Homogenates were then centrifuged at low speed to pellet nuclei and cell debris (3,000 g for 20 min at 4 °C) and the supernatant (S1) was then centrifuged at high speed (38,000 g for 30 min at 4 °C) to obtain the membrane fraction (P2 pellet). Membrane pellets were solubilized in cold RIPA buffer for

1 h at 4 °C and clarified by centrifugation (10 min, 15,000 g). Proteins were then extracted using methanol/chloroform precipitation and stored at −80 °C prior to mass spectrometry.

## Mass spectrometry sample preparation

Membrane fraction (or IP samples) were resuspended with 100 µl of 8 M urea for 30 min. Then 100 µl of 0.2% ProteaseMAX (Promega, Cat# V2072) was added and incubated for 2 h. The samples were reduced by TECP for 1 h and alkylated by IAA for 20 min at room temperature at dark. After diluting with 300 µl of 50 mM ammonium bicarbonate, 5 µl of 1% ProteaseMAX and trypsin (1:100) was then added for overnight incubation at 37 °C with intensive agitation. The next day, the reaction was quenched by adding 1% formic acid. Membrane fractionation samples were desalted using HyperSep C18 Cartridges (Thermo Fisher Scientific) and vacuum centrifuged to dry. The desalted samples were fractionated using Hypersep SCX columns (Thermo Fisher Scientific, Cat# 60108-420). Fractions were eluted twice in 300 µl buffer at increasing ammonium acetate concentrations (0, 25, 50, 500, 1000 mM), speed vacuumed to dryness then desalted by ZipTips (Pierce, Cat# 87784) and again dried down for a second time.

## Mass spectrometry

Three micrograms of each fraction or sample were auto-sampler loaded with a Thermo EASY nLC 1000 UPLC pump onto a vented Acclaim Pepmap 100, 75 µm × 2 cm, nanoViper trap column coupled to a nanoViper analytical column (Thermo Fisher Scientific, Cat#: 164570, 3 µm, 100 Å, C18, 0.075 mm, 500 mm) with stainless steel emitter tip assembled on the Nanospray Flex Ion Source with a spray voltage of 2000 V. A coupled Orbitrap Fusion (Thermo Fisher Scientific) with Xcalibur 4.4 was used to generate mass spectrometry (MS) data. Buffer A contained 94.785% H2O with 5% ACN and 0.125% FA, and buffer B contained 99.875% ACN with 0.125% FA. The chromatographic run was for 4 h in total with the following profile: 0–7% for 7, 10% for 6, 25% for 160, 33% for 40, 50% for 7, 95% for 5 and again 95% for 15 min receptively. CID-MS2 method was used as previously described ([79]). Briefly, ion transfer tube temp = 300 °C, Easy-IC internal mass calibration, default charge state = 2 and cycle time = 3 s. Detector type set to Orbitrap, with 60 K resolution, with wide quad isolation, mass range = normal, scan range = 300–1500 m/z, max injection time = 50 ms, AGC target = 200,000, microscans = 1, S-lens RF Level = 60, without source fragmentation, and datatype = positive and centroid. MIPS was set as on, included charge states = 2–6 (reject unassigned). Dynamic exclusion enabled with $n = 1$ for 30 s and 45 s exclusion duration at 10 ppm for high and low. Precursor selection decision = most intense, top 20, isolation window = 1.6, scan range = auto normal, first mass = 110, collision energy 30%, CID, Detector Type = ion trap, OT Resolution = 30 K, IT scan rate = rapid, max injection time = 75 ms, AGC target = 10,000, Q = 0.25, inject ions for all available parallelizable time.

## Mass spectrometry data analysis and quantification

Protein identification/quantification and analysis were performed with Integrated Proteomics Pipeline - IP2 (Integrated Proteomics Applications, Inc., San Diego, CA. http://www.integratedproteomics.com/) using ProLuCID[80,81], DTASelect2[82,83], Census and Quantitative Analysis. Spectrum raw files were extracted into MS1, MS2 files using RawConverter 1.0.0.0 (http://fields.scripps.edu/downloads.php). The tandem mass spectra were searched against UniProt mouse protein database (downloaded on 03-25-2014)[84] and matched to sequences using the ProLuCID/SEQUEST algorithm (ProLuCID version 3.1) with 20 ppm peptide mass tolerance for precursor ions and 600 ppm for fragment ions. The search space included all fully and half-tryptic peptide candidates within the mass tolerance window with no-miscleavage constraint, assembled, and filtered with DTASelect2

through IP2. To estimate peptide probabilities and false-discovery rates (FDR) accurately, we used a target/decoy database containing the reversed sequences of all the proteins appended to the target database[85]. Each protein identified was required to have a minimum of one peptide of minimal length of six amino acid residues; however, this peptide had to be an excellent match with an FDR < 0.001 and at least one excellent peptide match. After the peptide/spectrum matches were filtered, we estimated that the protein FDRs were ≤1% for each sample analysis. Resulting protein lists include subset proteins to allow for consideration of all possible protein forms implicated by a given peptide identified from the complex IE protein mixtures. Then, we used Census and Quantitative Analysis in IP2 for $^{15}$N-based protein quantification. Peptide ion intensity ratios for each pair of extracted ion chromatograms was calculated in Census. In brief, a linear least-squares correlation algorithm was used to define the ratio (i.e., line slope) and fitting quality [i.e., correlation coefficient (r)] between the data points of the unlabeled and labeled ion chromatograms. In this study, only peptide ratios with a $r^2$ greater than 0.5 were considered as reliable measurements and used for further analysis. We further used the Grubbs test to define statistical outliers. The cutoff is $p = 0.01$. Any data with a $p$ value greater than this threshold were removed from further analysis

## Z-ratio calculations

Z-scores was calculated by following formula:

$$Z_{(P,i)} = (P_i - Mean_i)/SD_i \qquad (1)$$

$Z_{(P, i)}$ designated as the Z-score of protein P in biological replicate (BR) i. $P_i$ designated as the value of protein P in BR i. $Mean_i$ designated as the mean value of BR i. $SD_i$ designated as the standard deviation of BR i.

Z ratio was calculated by following formula:[86]

$$Z\ ratio_p = [Z_{(p,dup/+)\_Avg} - Z_{(p,+/+)\_Avg}]/SD[Z_{diff(p1...pn)}] \qquad (2)$$

Z ratio$_p$ represents the Z ratio of protein P. $Z_{(p, dup/+)\_Avg}$ represents the average of all z scores of protein P acquired from 16p11.2$^{dup/+}$ mice. $Z_{(p, +/+)\_Avg}$ represents the average of all z scores of protein P acquired from 16p11.2$^{+/+}$ mice. $SD[Z_{diff(p1...pn)}]$ represents the standard deviation of all the Z score differences between 16p11.2$^{dup/+}$ and 16p11.2$^{+/+}$ mice for all the quantified proteins, $p_1....p_n$.

## Gene ontology analysis

Gene ontology analysis was performed using DAVID (https://david.ncifcrf.gov/) and SynGO (https://www.syngoportal.org/)[87,88]. For GO analysis using DAVID, all gene lists were converted to Entrez Gene IDs using DAVID Gene ID converter, generating a background membrane proteome list ($n = 4351$), an upregulated protein list ($n = 647$), and downregulated proteins list ($n = 1010$). Each list was compared to the background membrane proteome for enriched biological processes using the GOTERM_BP_4 function and Functional Annotation Clustering (Fig. 1c, g, and S6b). When using Functional Annotation Clustering a representative term of that cluster was presented and the full datasets are presented in supplementary data (Supplementary Data 2, 4, 6). Only Benjamini–Hochberg (BH) adjusted $p$-values were used in all results. Data are displayed as the -Log$_{10}$(BH-adjusted $p$-value). For GO analysis using SynGO, all lists were converted to HGNC IDs. Results are reported as -Log$_{10}$(FDR-corrected $p$-values).

## Gene set enrichment analysis

Mass spectrometry data was filtered to only include proteins that were quantified in at least three 16p11.2$^{+/+}$ and three 16p11.2$^{dup/+}$ samples. All dysregulated proteins (upregulated, Z-ratio>1.96 or downregulated, z-ratio < −1.96) in 16p11.2$^{dup/+}$ membrane fractions were considered in the

gene set enrichment analysis including (by gene symbol) 659 upregulated and 1024 downregulated proteins. Enrichment analysis was performed using a hypergeometric test implemented in R and custom genes sets. All genes/proteins that were not present in the background membrane proteome (i.e. all 4447 proteins detected in membrane fractions) were excluded from the analysis. Importantly, to control for any pre-existing enrichment of gene sets in membranes, all hypergeometric tests were performed with the membrane proteome as the background set. Data are presented as the % increase in enrichment over what is expected by chance with the associated hypergeometric $p$-value.

### Custom gene lists
Lists of genes containing de novo single nucleotide variants (dnSNVs) identified in individuals with neuropsychiatric disorders were obtained from previously published data compiled in the de novo database (http://denovo-db.gs.washington.edu/denovo-db) from large-scale exome sequencing studies[89–93]. Only dnSNVs predicted to affect protein coding were used in downstream analysis (missense, frameshift, stop gained, stop lost, start lost, splice acceptor, and donor variants). Variants outside coding regions, as well as synonymous variants were excluded. Control dnSNV lists were obtained from the same sources and include exonic variants from unaffected siblings or individuals with no known mental disorder.

The OMIM database was downloaded and gene lists associated with epilepsy or intellectual disability were generated by searching for key words using the GeneMap function. Epilepsy search terms included ("epilepsy", "epileptic", "seizure" and "seizures") and intellectual disability search terms included ("intellectual disability", "mental retardation" and "developmental delay").

Cell type specific genes set (astrocytes, neurons, oligodendrocytes) and excitatory/inhibitory genes sets were obtained from published supplementary material[94,95]. Genes enriched in excitatory neurons were defined as all genes expressed in neocortex with >2 transcripts per million (TPM) and a mean expression of 1.2 fold over inhibitory cell types (PV + and VIP + ) with $p < 0.05$. The same strategy was used to determine inhibitory gene sets, where enrichment over excitatory neuron expression was calculated.

Human brain co-expression gene sets were used to identify molecular networks functionally related genes in the 16p11.2 region[43]. The top 2000 co-expressed genes ($r^2$) for each gene was downloaded from the brain span portal (http://brainspan.org/rnaseq/search/index.html), selecting all human cortical regions and all stages of development.

Lists of disrupted proteins from human NPDs and mouse models were obtained from published proteomic datasets[96–102]. All proteins with $P < 0.05$ were included for the gene set analysis (Supplementary Fig. 3).

The PRRT2 interactome list contained all proteins from the IAP-MS experiment ($n = 208$) that passed the selection criteria ($P < 0.1$) (see methods section Immunoaffinity purification of PRRT2).

Lists of differentially expressed genes in cortex from 16p11.2[dup/+] mice (used at $P < 0.01$) were obtained from previously published studies[30,39].

### Protein:protein interaction network analysis
To generate a comprehensive epilepsy-associated network, we generated an expanded list of epilepsy risk factors (see Gene set enrichment analysis for details) including OMIM epilepsy-associated genes[103], epilepsy dnSNV data from the denovo database[90], supplemental data in ref. [89]. and recurrently affected genes with at least two dnSNVs in ref. [104]. The expanded epilepsy list was intersected with all the dysregulated proteins identified in the 16p11.2[dup/+] membrane proteome, which resulted in an overlap of 110 proteins. A list of gene symbols corresponding to each protein in the list were imported into the

GeneMANIA plugin within Cystoscape (v3.6.0). The network was built with automatic weighting, exclusively using physical interactions as edges, and finding the top 50 related genes and at most 20 attributes for each node. To incorporate newly identified PRRT2 interactions into the epilepsy subnetwork, a list of binary PRRT2 protein:protein interactions from the IAP-MS experiment was imported into Cystoscope as a network table, and subsequently merged with the epilepsy subnetwork. Cytoscape was used to visualize and annotate the network. Values of degree ($D$) and betweenness centrality ($C_B$) were calculated using Cytoscape's integrated network analysis module.

### Primary cortical neuron cultures and transfections
All experiments were performed on 17-23 day in vitro (DIV) primary mouse cortical neurons. Primary cortical neurons were prepared from dissociated sibling 16p11.2[dup/+] and 16p11.2[+/+] embryos at embryonic day 18.5. Brains were extracted from embryos, and cortices were isolated using a dissection microscope and stored in ice cold L-15 Medium (Thermo Fisher) supplemented with penicillin (50 U/ml) and streptomycin (50 μg/ml) (pen/strep, Gibco). The cortical tissue was digested in papain (Sigma; 20 units/ml) diluted in Dulbecco's Modified Eagle Medium (DMEM, Corning #10013CV) containing EDTA (0.5 mM, Sigma) and DNaseI (2 units/mL, Sigma). Papain solution was activated 10 min prior to enzymatic digestion with L-cysteine (1 mM) and incubated at 37 °C for 20 min. Cortical tissue was mechanically dissociated in high glucose dissociating medium (DMEM containing 10% Fetal bovine serum (FBS, Corning), 1.4 mM GlutaMAX, and 6 g/L glucose). The supernatant containing dissociated cortical neurons was seeded on 12 or 24 well plates containing coverslips (Neuvitro Corporation) precoated overnight with 50 μg/ml poly-D-lysine (Sigma). Twelve-well plates contained 18 mm coverslips seeded at a density of 200'000 cells per well and 24 well plates contained 12 mm coverslips seeded at a density of 100'000 cells per well. Two hours after seeding the dissociation medium was replaced with Neurobasal feeding medium (Neurobasal medium supplemented with B-27 supplement (Gibco), 2 mM GlutaMAX (Gibco) and pen/strep). Neuronal cultures were maintained at 37 °C in 5% $CO_2$ and half media changes of Neurobasal feeding medium were performed twice a week.

### Calcium imaging in cultured neurons
All experiments were performed at DIV19-23 primary cortical neurons cultured on 18 mm coverslips by experimenters blind to genotype. Spontaneous activity data is from 4 independent cultures (16p11.2[+/+]= 14 brains, 22 coverslips; 16p11.2[dup/+]=11 brains, 20 coverslips) and data with bicuculline is from 2 independent cultures (16p11.2[+/+]=5 brain, 15 coverslips; 16p11.2[dup/+]=6 brains, 15 coverslips). Cells were incubated with a fluorometric calcium indicator (Cal520AM, 1:1000, AAT biotech) and glial cell dye (SR101, 1:1000, Sigma) directly in the culture medium for 30 min, before being wash three times in warm ACSF containing in mM: 125 NaCl, 2.5 KCl, 26.2 NaHCO3, 1, NaH2PO4, 11 glucose, 5 HEPES, 2.5 CaCl2 and 1.25 MgCl2. Coverslips containing neurons were incubated for 30 min at 37 °C in the imaging chamber (Warner Instruments, QR-41LP) and transferred to an OKOLAB stage-type incubator with 5% CO2 at 37 °C prior to imaging. Neuronal network activity was acquired with epifluorescence at 2 frames per second for 10 min using a Nikon C2 + microscope equipped with an ANDOR Zyla 4.2 sCMOS camera and x10 objective. Spontaneous neuronal activity was recorded in either ACSF or ACSF containing 30 μM bicuculline (Tocris) which was perfused into the recording chamber via a peristaltic pump (Gilson). Regions of interests (ROIs) corresponding to the neurons somas (~300 per coverslip) were manually traced in NIS-Elements (v4.20 Nikon) to obtain calcium traces over the 10 min recording. Each coverslip was background subtracted using an area of the coverslip with no neurons. The calcium signals were normalized to the local baseline generating F/F0 values and imported into MATLAB.

## Calcium imaging of acute brain slices

Experiments were performed ~2 week old 16p11.2$^{dup/+}$ and 16p11.2$^{+/+}$ mouse pups (postnatal day 12-16) from the same litter, by an experimenter blind to genotype. Data on somatosensory cortex was generated from of 4 independent litters (16p11.2$^{+/+}$=4 mice, 7 slices; 16p11.2$^{dup/+}$=4 mice, 8 slices) and data on visual cortex was generated from of 3 independent litters (16p11.2$^{+/+}$=2 mice, 7 slices; 16p11.2$^{dup/+}$= 4 mice, 10 slices). Mice were deeply anesthetized with isoflurane 3.5%, decapitated and their brain rapidly extracted and chilled in ice-cold "cutting" ACSF composed of (in mM): NaCl 92, KCl 2.5, NaH$_2$PO$_4$ 1.2, NaHCO$_3$ 30, HEPES 20, glucose 25, sodium ascorbate 5, sodium pyruvate 3, MgSO$_4$.7H$_2$O 10, CaCl$_2$.2H$_2$O 0.5, pH 7.3 and bubbled with 95% O$_2$, 5% CO$_2$. Brains were glued on a vibratome VT1000S (Leica) and cut to a thickness of 400 μm. Slices were kept at least 45 min at 33 °C in the "recovery" ACSF (similar to cutting ACSF with 2 mM MgSO$_4$ and CaCl$_2$) before incubation with the calcium sensor Cal520AM (AAT biotech) and the glial dye SR101 (Sigma) for 45 min. The layer 2/3 of the primary somatosensory cortex (S1) or primary visual cortex (V1) was acquired using a multiphoton laser-scanning microscope (A1R MP, Nikon) at 4.8 frames per second with a laser power of 20 mW at 820 nm. Slices were incubated at 33 °C during acquisition in "recording" ACSF (in mM): NaCl 124, KCl 2.5, NaH$_2$PO$_4$ 1.2, NaHCO$_3$ 24, HEPES 5, glucose 12.5, MgSO$_4$.7H$_2$O 1, CaCl$_2$.2H$_2$O 2, pH 7.3, 95% O$_2$, 5% CO$_2$. The first five minutes were acquired in control ACSF followed by 25 min of bath application of bicuculline (50 μM) inducing large synchronized neuronal events. ROIs corresponding to neuronal somas were automatically traced on ImageJ using a custom-made routine detecting the contour of elements measuring from 40 to 500 μm2 and presenting a z-score variation of the Cal520 signal equal or superior to 2.5. ROIs were manually inspected to remove artifacts and glial cells (on average, less than 5% of the automatic ROIs are deleted). Calcium signals were imported in MATLAB, normalized using local background and smoothed with a wavelet filter, the calcium transients are expressed as the percentage of variation around the baseline (ΔF/F0).

## Calcium imaging analysis

Calcium events were automatically detected using the "findpeaks" function of the software with a minimum of 15% variation above the baseline (or 20% for acute slices). Peak amplitude, half-width and frequency was calculated within MATLAB. Calcium event amplitude was measured by subtracting the local baseline (F0) to the maximum peak value (F) and was normalized to a percentage by dividing by the local baseline: (F-F0)/F0. A Monte-Carlo simulation using 1000 permutations of the peaks for each neuron was used to unbiasedly detect the minimal number of co-active cells that could be considered a synchronized calcium event (threshold equal to the 99.9 percentile of the sum of events for the 1000 simulations). Following the detection of synchronized events, we determined the number of co-active cells per event and the frequency of network events within MATLAB. Neuronal synchronization was also assessed using a pairwise-correlation that measures the R coefficient correlation (a value of 1 being equal to a perfect positive correlation i.e. perfect synchrony) for the signals of all pairs of neurons. The delay between neurons is quantified by measuring the time separating a calcium event from the closest calcium event for every pair of neurons in an interval of plus or minus 2.1 s.

## Glutamate imaging in cultured neurons

Experiments were performed at DIV17-22 in primary cortical neurons. Data was collected from 5 independent cultures (WT = 6 brains, 20 FOV; Dup = 11 brains, 42 FOV; Corr=10 brains, 27 FOV). Neurons were plated on 18 mm coverslips (3 × 10$^5$/coverslip), infected with SyniGluSnFR virus (Addgene #98929-AAV1) at DIV11-14, and incubated at 37 °C and 5 % CO$_2$ for 6–8 days. Prior to imaging, coverslips were washed three times in warm ACSF (the same composition as for in vitro Ca$^{2+}$ imaging), incubated for 10 min at 37 °C directly in the imaging

chamber, and mounted in an OKOLAB stage-type incubator with 5% CO$_2$ at 37 °C. Spontaneous network activity was acquired with epifluorescence at 20 frames per second for 4 mins using a Nikon C2 + microscope equipped with an ANDOR Zyla 4.2 sCMOS camera and 10x objective. 1-2 fields of view (FOV) were acquired per coverslip. Fluorescence changes in the whole field of view were quantified in NIS-Elements (v4.20 Nikon) and traces were imported into MATLAB for quantification. The baseline of the signal was normalized using the "medfilt1" function and peaks were detected using "findpeaks", with a minimum of 1% variation above the baseline. Evoked peaks were elicited by adding KCl (30 mM final conc.) directly into the ACSF of the imaging chamber. KCl-induced peaks were normalized by dividing fluorescence values by the baseline fluorescence (average of the first 50 frames). Normalized peaks were imported into GraphPad Prism. The Area Under Curve (AUC) function was used to calculate amplitude and AUC of the peaks, whilst a nonlinear fit was used to estimate the decay portion of the curve (T$_{1/2}$).

## Immunoaffinity purification of PRRT2

For each immunoaffinity purification (IAP) experiment ($n = 3$), 10 μg of PRRT2 rabbit antibody (Sigma, HPA014447) and non-specific rabbit IgG antibody (Santa Cruz, sc-2027) was crosslinked to 30 μl of packed Protein A/G beads (Pierce # 20421). Antibodies and beads were coupled in phosphate buffered saline (PBS), followed by three washes in 0.2 M sodium borate, pH 9. Antibodies were crosslinked using 20 mM dimethyl pimelimidate (DMP, Thermo Fisher) in sodium borate buffer. Coupled beads were washed once in 0.2 M ethanolamine (pH 8.0) and the remaining DMP was quenched in ethanolamine for 1 h at RT. Beads were washed three times in PBS and stored at 4 °C until the IAP experiment (<2 days). Washed membrane fractions were prepared essentially as previously described[105]. For each IAP experiment, the neocortex from 8–10 six-week-old mice was homogenized in cold sucrose buffer (20 mM HEPES pH 7.4, 320 mM sucrose, 5 mM EDTA) supplemented with protease inhibitor cocktail (Roche). Homogenates were centrifuged at low speed to pellet nuclei and cell debris (3,000 g for 20 min at 4 °C) and the supernatant (S1) was then centrifuged at high speed (38,000 $g$ for 30 min at 4 °C) to obtain a membrane pellet (P2). P2 was re-suspended in potassium iodide buffer (20 mM HEPES pH 7.4, 1 M KI, 5 mM EDTA) to remove membrane-associated proteins (S3), and membranes were again collected by centrifugation (38,000 $g$ for 20 min at 4 C). Membranes were washed (20 mM HEPES pH 7.4, 5 mM EDTA) and pelleted once more (S4) before solubilizing in CHAPS buffer (20 mM HEPES pH 7.4, 100 mM NaCl, 5 mM EDTA, 1% CHAPS) supplemented with protease inhibitors for 2 h at 4 °C. Solubilized membranes were clarified by centrifugation for 30 min at 100'000 $g$, 4 °C and the clarified supernatant retained for the IAP experiment (S5). The CHAPS-insoluble pellet was re-suspended in SDS buffer (50 mM TRIS pH 7.4, 150 mM NaCl, 1% SDS) supplemented with protease inhibitors, solubilized at 37 °C for 20 min and clarified by centrifugation (S6). All remaining fractions were solubilized in CHAPS buffer. The washed membrane fraction (S5) was incubated with crosslinked antibody beads overnight at 4 °C and the next day, beads were collected in purification columns (Biorad). Beads were washed three times in CHAPS buffer before eluting protein complexes in 100 mM glycine pH 2.5 containing 1% CHAPS. Eluate was then neutralized with 1 M Tris pH 8.5 (1:10) and proteins extracted using trichloroacetic acid (TCA) precipitation. Precipitated proteins were stored at −80 °C prior to MS. Common MS contaminants were removed. All proteins that showed a mean spectral count that was 2 fold higher in the PRRT2 IAP compared to IgG (one-tailed $t$-test, $p < 0.1$) were considered part of the interactome dataset ($n = 208$).

## Immunoprecipitation in HEK cells and mouse neocortex

HEK293T cells (ATCC) were cultured in DMEM containing 10% FBS and 1% pen/step. Cells were seeded onto 10 cm plates and transfected with

lipofectamine 2000 using manufacturer's instructions. 5 µg of FLAG-*Prrt2* plasmid (GeneCopoeia) and 5 µg of SNARE plasmid (T7-VAMP2 or myc-STX1A, kindly provided by Mitsunori Fukuda, Tohoku University) were transfected and allowed to express for 48 h prior to immuno-precipitation (IP). HEK293T cells were washed with PBS, homogenized in IP buffer (50 mM Tris, pH 7.4, 150 mM NaCl, 1% Triton X-100, with protease inhibitor cocktail) and solubilized for 1 h at 4 °C. Solubilized material was clarified by centrifugation at 15,000 g for 10 min at 4 °C. Soluble proteins were then incubated with 1 µg of anti-FLAG antibody (F1804, Sigma) or 1 µg of mouse IgG (sc-2025, Santa Cruz) overnight at 4 °C, followed by a 1 h capture with 20 µl protein A/G beads the following day. Beads were then washed three times with IP buffer before adding 2× Laemmli buffer (Biorad) and boiling at 95 °C for 5 min. Mouse brain IPs were performed in adult mice (>6 weeks), essentially as above. The sodium channel IP was performed on a solubilized P2 membrane fraction from neocortex, using IP buffer and 5 µg of anti-pan-Na$_v$ antibody (S8809, Sigma). Input and IP Samples were analyzed by SDS-PAGE and western blotting.

### Seizure susceptibility assays

All experiments were performed on mice 6–8 weeks of age, using littermate controls and by experimenters blind to genotype. Initial experiments (Fig. 3) were performed on 3 independent cohorts (WT, n = 16 mice; Dup, n = 17 mice). Rescue experiments (Fig. 6) were performed on 5 independent cohorts (WT, n = 14 mice; Dup, n = 13 mice; Corr, n = 16 mice; Het, n = 16 mice). Mice were placed in a holding room at least 24 h prior to experimentation. The following day, mice were placed in a clear, plexiglass chamber and their pre-induction behavior were recorded 15–30 min. Mice were then injected with 28 mg/kg of kainic acid (Sigma) intraperitoneally, a known chemo-convulsant used to assess seizure susceptibility in mice. Post-injection behavior was recorded for up to two hours and was ceased after the death of the animal. Behavioral assessments were performed by experimenters blind to genotype. The primary outcome measures in our analysis were time of onset of generalized tonic-clonic seizures (GTCS) and time of death. Additionally, mice were monitored and scored at 1 h post-injection for seizure related behaviors according to a modified Racine scale[106] as follows: 1. Immobility. The mouse lays flat on the ground without moving. 2. Head nodding and signs of rigidity (erect tail, stretched out forelimbs) 3. Forelimb clonus. Involuntary muscle contractions in the forelimbs. 4. Dorsal extension ("rearing") and forelimb clonus. The mouse has a considerable loss of balance combined with clonus. 5. Persistent rearing and falling. The animal may be also be rolling around repeatedly on the ground ("barrel rolling"), have a brief seizure spell (<1 s) or running around intensely. 6. Generalized tonic-clonic seizure. The mouse is on the ground unable to right itself with convulsive tonic and clonic muscle contractions for at least 5 s. 7. Death of the animal due to severity of GTCS. The data is presented as Kaplan–Meier curves which represent cumulative percentages over time. The percentage change in latency to GTCS (Supplementary Fig. 8) was calculated by dividing the mean latency for 16p11.2$^{corr}$ to the mean latency for 16p11.2$^{dup/+}$, for each seizure induction trial where both genotypes had a GTCS (n = 11).

### Mouse behavior

All experiments were performed on mice at 6–10 weeks of age, blind to genotype. Open field test: Open filed tests were performed on 3 independent cohorts (WT, n = 20 mice; Dup, n = 14 mice; Corr, n = 17 mice; Het, n = 13 mice) Mice were placed in an arena (30 cm × 30 cm x 30 cm) and locomotor activity was recorded using an overhead camera. Mice were monitored for 30 min and tracking software (Ethovision, Noldus) was used to assess the total distance travelled. 3-chamber social interaction test: Sociability tests were performed 3 independent cohorts (WT, n = 17 mice; Dup, n = 7 mice; Corr, n = 16 mice; Het, n = 12 mice) in a rectangular glass arena with divided into 3 chambers, with a single entry between each chamber. The test was composed of two phases: In the first phase (habituation), object (inverted wire cup) was positioned in each of the left and right chambers. Mice were placed in the central chamber and allowed to explore the arena for 5 min. In the second phase (Sociability), a novel mouse (of same sex and unknown to the test mouse) was placed under one of the cups, whilst the other cup remained empty. The test mouse was then placed into the middle chamber and allowed to explore for 5 min. The amount of time each mouse spent investigating the empty object or novel mouse (defined as sniffing or interacting) was manually recorded. The data for each mouse was converted to a discrimination index (DI (%) = [(t$_{mouse}$-t$_{object}$)/ (t$_{mouse}$ + t$_{object}$)×100]).

### Statistical analysis

All statistical tests were performed with GraphPad Prism or R. All data was tested for normality using the D'Agostino and Pearson omnibus normality test. An unpaired t-test was used if the data was parametric and a Mann-Whitney test was used if the data was non-parametric. Bar graphs are displayed as mean ± SEM, unless otherwise specified. For seizure susceptibility assays, survival and GTCS onset curves were analyzed using Log-rank test. Percentage survival and GTCS was also compared using Fisher's exact test. A post-hoc Bonferroni correction was then performed to determine the statistical significance of at each distance interval. For behavioral with Corr mice, a two-way ANOVA with Tukey's post-hoc test was performed with effect of *Prrt2* genotype and effect of 16p11.2 genotype as independent variables. For glutamate imaging, a Kruskal–Wallis test with Dunn's correction for multiple comparison was performed. A single outlier was removed based on a Grubb's test at α=0.01, when comparing amplitude. Hypergeometric tests for gene set analysis were performed in R. p-values were considered significant if p < 0.05, unless otherwise specified. For all statistical tests asterisks indicate: *p < 0.05; **p < 0.01, ***p < 0.001. The "n = " for each experiment refers to either: the number of cells (for neuron morphology, spine analysis and puncta analysis), the number of coverslips (for calcium imaging in vitro), the number of slices (for acute brain slices), the number of FOV (for glutamate imaging) or number of animals (for seizure and behavioral assays).

### Reporting summary

Further information on research design is available in the Nature Portfolio Reporting Summary linked to this article.

## Data availability

The 16p11.2$^{dup/+}$ membrane proteome and PRRT2 IP-MS data generated in this study have been deposited in the MassIVE (MSV000090884) and ProteomeXchange (PXD038753) databases. The PRRT2 interactome has also been deposited in BioGRID and an annotated version is available in the Supplementary Information. Source data are provided with this paper.

## Code availability

The code used to analyze Ca$^{2+}$ and glutamate imaging data, along with example traces, are freely available for download at https://github.com/marcdossantosPHD/PenzesLabimaging

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

## Acknowledgements

We would like to thank Professor Nicolas Katsanis (Duke University) for providing the backcrossed 16p11.2^dup/+ mice and Professor Mitsunori Fukuda (Tohoku University) for the T7-VAMP2 and STX1A mouse cDNA plasmids. *Prrt2*+/- mice were kindly provided by the University of Tennessee Health Science Center. We are very grateful to the Behavioral Phenotyping Core at Northwestern University for the use of their equipment and expertise. This work was supported by National Institute of Mental Health (NIMH) and National Institute of Neurological Disorders and Stroke (NINDS) Grants: MH097216 (to P.P.), NS108874 (to A.L.G.), NS053792 and NS084959 (to J.A.K.), R56NS094965, R03NS101485 and R56NS123059 (to M.S.L.), an Individual Biomedical Research Award from The Hartwell Foundation (J.N.S.) and a Swiss National Science Foundation Early Postdoc Mobility Fellowship (P2SKP3_161675 to M.P.F.).

## Author contributions

M.P.F., A.L.G., J.A.K., J.N.S. and P.P. designed research. M.P.F., M.D., N.H.P., Y.Z.W., D.S., N.A.H., M.D.M-S., R.G., S.Y., M.S.L., and K.E.H., performed research. M.P.F., N.H.P., Y.Z.W., D.S., M.D., L.E.D., N.A.H., and V.A.B., analyzed data. M.S.L. generated and provided mice, and M.P.F. and P.P. wrote the manuscript.

## Competing interests

The authors declare no competing interests.

## Additional information

[1]Department of Neuroscience, Northwestern University Feinberg School of Medicine, Chicago, IL 60611, USA. [2]Center for Autism and Neurodevelopment, Northwestern University Feinberg School of Medicine, Chicago, IL 60611, USA. [3]Department of Neurology, Northwestern University Feinberg School of Medicine, Chicago, IL 60611, USA. [4]Department of Pharmacology Northwestern University Feinberg School of Medicine, Chicago, IL 60611, USA. [5]Department of Psychology, University of Memphis, Memphis, TN 38152, USA. [6]Veracity Neuroscience LLC, Memphis, TN 38157, USA. [7]Present address: Instituto Universitario de Investigación en Neuroquímica, Departamento de Bioquímica y Biología Molecular, Facultad de Farmacia, Universidad Complutense, 28040 Madrid, Spain. ✉e-mail: p-penzes@northwestern.edu

