## [Peer Review File · Nature Communications]

Rescue of neuropsychiatric phenotypes in a mouse model of
16p11.2 duplication syndrome by genetic correction of an
epilepsy network hubEditorial Note: Parts of this Peer Review File have been redacted as indicated to maintain the confidentiality of unpublished data.

REVIEWER COMMENTS

Reviewer #1 (Remarks to the Author):

Review Nature communications

This is an exciting piece of work by Forrest et al, corresponding author Penzes. Using the 16p11.2 duplication mouse model, the authors investigate the proteome of the synaptic membrane in neocortex and identify a network relevant to epilepsy, where the PRRT2 protein is a hub. They also identify cellular and circuitry based dysfunction, and produce a robust seizure phenotype in response to a chemical stimulus. Restoring PRRT2 to WT levels ameliorates the seizure phenotype and also improves social function in the mice, suggesting PRRT2 is indeed a relevant contributor to the seizure phenotype. This is a strong paper that will add to the literature about CNV disorders generally, and 16p11.2 specifically. However, the authors have been too expansive in their interpretation of the data. I do not believe the data in the paper support the conclusion that PRRT2 is relevant to all comorbid NPD phenotypes associated with the 16p11.2 duplication. Specific comments follow:

Assuming this paper is relevant to the 16p11.2 BP4-BP5 region. The authors should be specific.

Introduction, page 3, line 60: should be “increase risk FOR multiple disorders...”

Intro page 3 line 62-63, what does it mean to be a “leading CNV in SZ and ASD”?

General comment: In human genetics “microduplication” and “microdeletion” have fallen out of favor, it is preferred to use “duplication” or “deletion”

Page 4 lines 76-77 – deficits in social behaviors that are exhibited by laboratory mice can in no way be considered an endophenotype of human autism or schizophrenia (PMID: 33957550).

The introduction is about SZ and ASD but the findings are most relevant to epilepsy. This intro feels like a bait-and-switch.

They studies neocortex, but 16p11 is documented to have changes in the thalamus, insula, ventral striatum, and other places (pmid PMID: 25421402). The study of neocortex is poorly justified.

Page 6, last paragraph: to assess the genetic risk for NPD phenotypes, it is appropriate to do a formal meta-analysis, where a comprehensive set of search terms for each phenotype are delineated and the literature is thoroughly and systematically assessed. The procedure used appears to be quite ad-hoc and the manuscripts used to formulate the effect sizes seem to be cherry-picked. Furthermore, many papers (such as D’Angelo et al 2016) distinguish between probands and non-proband carriers, as the impact of the duplication is quite different between these two groups. This analysis is a weakness of the paper.

Page 7, line 176: this paragraph seems to advocate for the generalizability of the proteomics finding: a protein network associated with epilepsy was found to be dysregulated in the membrane fraction of the neocortex. When compared to other mouse models of NPDs such as FMR1, shank3, and others, they find common targets and argue that these proteins must therefore be relevant the autism and schizophrenia pathology. This logic is flawed, as seizures/epilepsy are common in all of the comparator disorders. The authors cannot distinguish whether they have identified a network that is general to NPDs globally, or specific to epilepsy. 12% of fragile X patients suffer from seizures, <https://doi.org/10.3389/fped.2021.736255>; 12-27% of SHANK3 subjects have epilepsy, PMID: 27554343; 11% in 22q11 PMID: 30977115. This

inference is also present in the discussion (page 17 line 448). An alternative conclusion is that the authors have identified targets and pathways relevant to the epilepsy/seizure phenotypes, and these may or may not be independent from SZ and ASD phenotypes. The balance of data in the paper – specifically, one behavioral test in mice with minimal face validity – does not support the expansive conclusions in the paper. The overlap with SZ and ASD postmortem tissue is again not conclusive, given the comorbidity across these disorders. There is nothing here to distinguish whether the protein network the authors have discovered is specific to seizures or relevant to a wider array of NPD phenotypes.

Page 12, line 311: This paragraph is somewhat confusing. The authors identify membrane proteins that are changed when the entire 16p11 region is duplicated. They then identify proteins that interact with a single gene in the region, PRRT2. It seems that overlapping targets contained in these two groups is to be expected? Why would this qualify as “surprising overlap” (line 319)? If PRRT2 is disrupted in the 16p11.2 duplication, wouldn’t one expect this to encompass the entire PRRT2 proteome? The fact that only 72% of the PRRT2 interactome is disrupted in the larger 16p11 dup proteome seems to be less than what is expected, not more. I would want to know the numbers in the other direction: how much of the entire 16p11 dup disrupted proteome is accounted for by duplication of PRRT2 alone? Those data might demonstrate that PRRT2 is in fact a driver of the underlying biological perturbations.

Page 14, line 382: correction of the induced seizure phenotype is a very strong finding and the most rigorous evidence in the paper that PRRT2 may drive seizure phenotypes. However, the reversal of the social deficit phenotype, while encouraging, cannot reasonably be interpreted as a marker of all NPD phenotypes, nor can it be interpreted as a “core behavioral endophenotype associated with ASD and SZ.” Other phenotypes associated with SZ and ASD have not been tested such as marble burying (stereotypy), vocalization frequency (deficits in language), and PPI (especially relevant for SZ). Furthermore, no learning and memory phenotypes were tested. I do not believe that this paper conclusively demonstrates that “correction of PRRT2 dosage can rescue core neuropsychiatric domains” in 16p11.2 duplication mice.

It is somewhat surprising that the authors compared their proteomic data to other mouse models and human datasets, but do not present the overlap between the 16p11.2 dup dysregulated proteome and the transcriptome, of which there are many available datasets.

Reviewer #2 (Remarks to the Author):

Human chromosome 16p11.2 is one of the recurrent CNV loci, including deletion and duplication in ASD, SZ, or ID individuals. The authors conducted quantitative proteomics analysis followed by bioinformatics approaches to identify a driver gene or dysregulated network in the brain of the 16p11.2 duplication mouse model. They found that an epilepsy-associated protein subnetwork was especially dysregulated in 16p11.2dup/+ mice. In addition, calcium imaging analyses by using primary cortical neurons or slices derived from 16p11.2 dup/+ mice showed enhanced functional connectivity or glutamate release. Next, the authors focused on the Prrt2 gene, which is likely to be the most reasonable gene for pathogenesis of 16p11.2 duplication based on their bioinformatics analyses. They performed the Prrt2 interactome and found that all known proteins that bind to Prrt2 are listed in the candidates. These candidates are again enriched in the epilepsy network, suggesting Prrt2 may have a central role in the epilepsy network in 16p11.2 genes. Finally, the authors corrected the copy number of Prrt2 in 16p11.2 by crossing with heterozygote of Prrt2 mice, named 16p11.2corr. This line showed a normalized level of seizure susceptibility or abnormal social interaction found in 16p11.2 dup/+ mice. Taken together, the authors concluded that the proteomics approach helps identify not only genes but also disease hubs in complex genetic disorders such as CNVs.

The authors' analyses or strategies are sophisticated and quantitative. So, the results seem to be

reliable and convincing conclusions. The reviewer has very few comments.

#1. Page 8, Fig. 2A-B: From this sentence, we cannot know whether these samples are primary neurons, slice cultured neurons, or in vivo neurons.

#2. Page 21: The genetic background of 16p11.2 dup/+ mice should be described clearly. After obtaining this line, how many times did the authors cross with C57BL/6N?

#3. As the authors previously showed, 16p11.2 dup/+ mice exhibited increased dendritic arborization (K D Blizinsky et al., 2016). This is an interesting view to see dendritic arborization in 16q11.2Crr mice.

Reviewer #3 (Remarks to the Author):

Here the authors describe a mouse model of CNV previously associated with various neurological disorders, the microduplication 16p11.2. Using this mouse model, they performed unbiased proteomic assays on membrane proteins in brain and identified a large amount of proteomic dysregulation, including networks involved in epilepsy, which were also associated with hypersynchronous activity in neurons and slices from 16p11.2 mutant mice. They also performed rescue experiments using a single gene, *Prrt2*, to reverse electrophysiological and behavioral, namely seizures, phenotypes. Overall, the manuscript was well-written and included important data and mechanistic insights into the seizure phenotype associated with this CNV. However, some major questions were left unanswered and the framing of the paper could be substantially improved.

Positive comments:

1. Use of unbiased proteomics on membrane proteins in this CNV mouse model is an important dataset for the field, especially because this CNV is associated with so many different phenotypes and disease outcomes in humans.
2. Calcium imaging experiments and behavioral phenotyping for seizures, including rescue of the phenotype with *Prrt2*, were fascinating and will be of great interest to the epilepsy/seizure field.

Major drawbacks:

1. The way the authors frame the manuscript, as this CNV being associated with shared risk for neuropsychiatric disease, especially ASD and SZ, seems out of place. Primarily, these two disorders seem cherry picked from the list of diseases associated with 16p11.2 microduplication, but then the entire paper focuses on epilepsy and seizure phenotypes, which is arguably a neurological disorder but not a neuropsychiatric disorder. The reviewer recommends the authors focus on the epilepsy phenotypes and not cloud the manuscript with less relevant information about shared risk with psychiatric disease, as their imaging and behavioral data only support the former. Further, Figure 1 shows various datasets the authors mined to create a link between their proteomics data and various neuropsychiatric/neurological diseases, and this seems unnecessary based on the later experiments.
2. Similarly, the inclusion of social behavior data in the last figure seems extremely out of place as well, as none of the mechanistic studies (which are very interesting in the context of seizures and epilepsy) can be directly related to social behavior, and these tests should not be included. Additionally, in line 77, the authors state that deficits in social behavior exhibited by 16p11.2 mice are common to SZ and ASD, and this is quite a stretch, as social deficits are involved in most psychiatric diseases and only a core symptom of ASD, not SZ.
3. The authors should include an additional experiment investigating the transcriptome of these mice using RNAseq. They state in the manuscript that transcriptomic outcomes are important for CNVs and this one in particular, yet do not include that data and jump straight to proteomics.
4. The above data (transcriptomics) would also be useful in the case of explaining the large number of proteomic changes reported that are NOT directly attributable to the genes in the genomic region

altered by this CNV. This should be further explained and investigated in the manuscript, as ~1600 differentially expressed proteins is very different from the 27 protein coding genes involved in that genomic locus.

5. There is no evidence of effect size or subject/cell numbers (N) in any of the manuscript or figures, and this is especially important in experiments where it is unclear if the data points on the graph represent one biological replicate (animal) or one cell (for example, all calcium imaging experiments)

5. The authors show an "increased burden of risk factors associated with many 16p11.2 related NPDs" in Fig 1 based on their proteomic data, but the data appears to basically include a large number of proteins involved in synaptic functioning that are generally impaired when brain disease is present. It may be better to focus on their neural transmission role and not necessarily attribute this to a specific disorder.

Other comments:

1. Why do the authors choose membrane proteins to screen compared to other subcellular fractions or total cell lysate? Can the authors clarify if the membrane fractions are essentially synaptosomes or if they are distinct in some way?
2. Line 179- membrane protein or total protein in referenced study?
3. In figure 2, it is not clear right away that these studies are done in neuronal cultures and not in vivo due to the wording of the figure caption and manuscript. It would be useful to distinguish this and to also distinguish it from the following data using slices instead of cells.
4. In Figure 2 A-B, a wild type comparison figure is needed.
5. Figure 6D, F- need legend to denote groups for each color on these graphs.

Reviewer #4 (Remarks to the Author):

A major problem in interpreting deletions and duplication of multigenic regions is untangling the role that each affected gene plays in producing the phenotype. There are many CNVs that produce neurological disorders and the 16p11.2dup/+ mice are an ideal model system. The authors have done an outstanding job in unravelling how the duplicated region produces complex phenotypes through the integration of many methods. The project exemplifies how proteomic approaches that leverage the wealth of publicly available bioinformatics can lead to the hypothesis that PRRT2 overexpression is a key cause of the phenotypes in 16p11.2dup/+ mice, which they substantiate using a genetic rescue approach. I recommend publishing the manuscript and suggest the following modifications to the text:

On page 6, the authors perform "gene set analysis to determine if the dysregulated proteins contained a burden of disease associated gene products identified by de novo exome sequencing studies of NPDs". They say "We found an increased burden of risk factors associated with many 16p11.2dup/+ related NPDs in our dataset (Fig. S1G and Fig. 1D). These data are reminiscent of the diverse clinical spectrum observed in CNV carriers and suggest that subnetworks relevant to each disease may be dysregulated in 16p11.2dup/+ mice." How do we know this doesn't just reflect an enrichment in the expression of NPDs in the brain? i.e. neural expressed proteins are going to be enriched in neural gene mutations. For this reason I do not agree that the following statement is surprising: "Surprisingly, the most enriched disease gene set was associated with epilepsy risk factors, followed by ID, SZ and ASD gene sets". The authors need to explain in the results section how they account for the bias in enrichment that results from brain gene expression. If they cannot do this, they need to tone down the text. I accept that they can present the data to make the link to epilepsy

Page 6, last paragraph, which reads "To determine the clinical relevance of these molecular disease-associations..." The authors should explain how they calculate the conferred risk. I also think it is very ambiguous to say "clinical relevance" because what is clinically relevant is to most doctors is how they treat patients. It would be more informative to give a more accurate description that aligns with the

analysis they present, which appears to be a ranking of diseases.

Page 7, the second paragraph says: “To determine the broader relevance of this network, and the global proteomic changes in 16p11.2dup/+ mice, we compared our datasets to disrupted proteins in a range of NPD mouse models, and human post-mortem proteomic studies of ASD and SZ (Fig. 1I). Gene set analysis revealed that a majority of mouse models we evaluated were enriched for global protein alterations observed in 16p11.2dup/+ mice (Fmr1^{-/y}, Shank3^{-/-}, 22q11.2^{+/-}, Cul3^{-/-}). Human datasets for ASD and SZ were remarkably, also enriched, indicating a convergence of human and mouse disease processes.”

What does it mean to say “we compared our datasets to disrupted proteins in a range of NPD mouse models, and human post-mortem proteomic studies of ASD and SZ” and “Human datasets for ASD and SZ were remarkably, also enriched”? For the reader to understand this they need to dig around in some supplementary information or methods section (without being directed there by the authors) and even then it is not obvious. This paragraph is vague and needs a better description.

Page 8, Section entitled “Enhanced functional connectivity etc”. There is no mention in this paragraph of whether they were looking at neurons in the intact mouse or cultures (the reader has to go to the figure legend). Again, a problem with clarity of description.

Figure 1B. chromosome is misspelled.

Seth Grant
Edinburgh University

Reviewer #1 (Remarks to the Author):

Review Nature communications

This is an exciting piece of work by Forrest et al, corresponding author Penzes. Using the 16p11.2 duplication mouse model, the authors investigate the proteome of the synaptic membrane in neocortex and identify a network relevant to epilepsy, where the PRRT2 protein is a hub. They also identify cellular and circuitry based dysfunction, and produce a robust seizure phenotype in response to a chemical stimulus. Restoring PRRT2 to WT levels ameliorates the seizure phenotype and also improves social function in the mice, suggesting PRRT2 is indeed a relevant contributor to the seizure phenotype. This is a strong paper that will add to the literature about CNV disorders generally, and 16p11.2 specifically. However, the authors have been too expansive in their interpretation of the data. I do not believe the data in the paper support the conclusion that PRRT2 is relevant to all comorbid NPD phenotypes associated with the 16p11.2 duplication. Specific comments follow:

Assuming this paper is relevant to the 16p11.2 BP4-BP5 region. The authors should be specific. This has been corrected in the introduction, thank you for pointing it out.

Introduction, page 3, line 60: should be “increase risk FOR multiple disorders...”
This, and all other instances, have been corrected

Intro page 3 line 62-63, what does it mean to be a “leading CNV in SZ and ASD”?
We have corrected the wording to be more specific in the text.

“Meta-analyses have identified this CNV as one of the most statistically robust variants associated with SZ and ASD (odd ratios 8-14), confirming initial studies”

General comment: In human genetics “microduplication” and “microdeletion” have fallen out of favor, it is preferred to use “duplication” or “deletion”.

We had not realized this was the case, thank you for bringing it to our attention. It has now been corrected throughout the manuscript.

Page 4 lines 76-77 – deficits in social behaviors that are exhibited by laboratory mice can in no way be considered an endophenotype of human autism or schizophrenia (PMID: 33957550).

Thank you for the excellent reference. We agree with the reviewer that we must take care to not anthropomorphize mouse behaviors. We do however still believe behavioral phenotyping can be used as part of a set of complementary techniques to assess deficits in mouse models. Hence, we have re-evaluated our interpretation of the mouse social behavior data throughout the manuscript. We have completely rewritten our introduction and discussion with this in mind, and have dedicated a paragraph in the discussion to address this important point.

“Despite this, care must be taken to draw parallels between human and mouse phenotypes. The 16p11.2^{dup/+} mouse model has strong construct validity because the same set of genes are duplicated in mice compared to human conditions. However, it is not possible to discern whether behavioral deficits exhibited in mice (face validity) are a true reflection of social impairments observed in NPDs. Sociability alterations in 16p11.2^{dup/+} mice could represent phenocopies of

human symptoms that are not mechanistically linked⁷⁶. Nevertheless, social deficits appear to be one of the most reproducible behavioral phenotypes in *16p11.2^{dup/+}* mice and are rescued correcting PRRT2 expression.”

The introduction is about SZ and ASD but the findings are most relevant to epilepsy. This intro feels like a bait-and-switch.

We agree with this reviewer that some of the central findings of this paper may seem most relevant to epilepsy (e.g. hypersynchrony and seizure susceptibility phenotypes). As such, we have substantially remodeled our introduction and discussion to focus on epilepsy as a co-morbid phenotype in NPDs. We feel this provides a fair focus on epilepsy without excluding the relevance this model has to other NPDs.

Our initial rationale for focusing the introduction on SZ and ASD was that 16p11.2 duplication has the strongest genetic association with SZ and ASD (most significant p-value, high odds ratios (8-14), and replication in large meta-analyses), whereas the CNV is not a general risk factor for epilepsies (PMID: 32568404). We set out our investigation expecting to find alterations in SZ and ASD-related proteins that would provide a direct link to SZ/ASD pathophysiology. Instead, we were surprised to find the most significant protein alterations were associated with epilepsy. This suggests that at the membrane proteome level at least, the proteomic modifications directly downstream of 16p11.2 genes are in epilepsy risk factors. We also would like to point out that the 16p11.2 dup mice have increased seizure susceptibility, not spontaneous seizures, which suggests a subclinical phenotype of network dysfunction and synchrony, which when combined with yet unknown factors, may lead to overt epilepsy (as is often seen in duplication carriers). We would therefore argue that focusing our manuscript solely on epilepsy would not accurately convey the complexity this CNV syndrome, given its strong association with ASD, SZ, and other NPDs.

Another aspect we would like to clarify is that we propose that altered network synchrony can be common substrates of both epileptic and some non-epileptic phenotypes (such as social behaviour). We have noted in the discussion that the regulation of neuronal synchrony is important for normal information processing in the cortex and deficits have been linked to multiple NPDs (aside from epilepsy). Thus, hypersynchrony would be predicted to impact both seizure susceptibility and also cortical functions linked to NPDs, as has been highlighted in animal models [PMID: 35290792 and 32123378]. We do regret if these concepts were not well articulated in the text and thus, they have now been clearly separated and explained in subsections of the discussion. Please see:

“Implications of circuit phenotypes for epilepsy” and “Implications of circuit phenotypes for other NPDs”

They studies neocortex, but 16p11 is documented to have changes in the thalamus, insula, ventral striatum, and other places (pmid PMID: 25421402). The study of neocortex is poorly justified.

We agree that the 16p11.2 duplication affects multiple brain structures and in no way do we conclude that the cortical deficits must be responsible for all the symptoms of CNV carriers. However, we did find it important to begin our study in the cortex as it has been implicated extensively in the etiology of NPDs through genetics and post-mortem studies. Future studies will

aim to assess alterations in other brain regions. To convey that other structures are likely involved in mediating disease, we have modified the following sentence in the text:

“The 16p11.2 duplication alters the structure of multiple brain regions in mice and humans. Here, we focused on studying proteomic dysfunction in the neocortex, a brain region that exhibits molecular and structural abnormalities in several NPDs”

Page 6, last paragraph: to assess the genetic risk for NPD phenotypes, it is appropriate to do a formal meta-analysis, where a comprehensive set of search terms for each phenotype are delineated and the literature is thoroughly and systematically assessed. The procedure used appears to be quite ad-hoc and the manuscripts used to formulate the effect sizes seem to be cherry-picked. Furthermore, many papers (such as D’Angelo et al 2016) distinguish between probands and non-proband carriers, as the impact of the duplication is quite different between these two groups. This analysis is a weakness of the paper.

The odds ratio calculations were based on studies that had the largest available sample size for each disorder. Because of differing CNV rates in the control samples, we selected the largest control sample available (PMID: 27773354) and used this frequency for all OR calculations, to meaningfully compare between disorders. We do accept that a formal meta-analysis would be a better way of comparing risk between illnesses and have therefore removed this panel from Figure 1.

Page 7, line 176: this paragraph seems to advocate for the generalizability of the proteomics finding: a protein network associated with epilepsy was found to be dysregulated in the membrane fraction of the neocortex. When compared to other mouse models of NPDs such as FMR1, shank3, and others, they find common targets and argue that these proteins must therefore be relevant to the autism and schizophrenia pathology. This logic is flawed, as seizures/epilepsy are common in all of the comparator disorders. The authors cannot distinguish whether they have identified a network that is general to NPDs globally, or specific to epilepsy. 12% of fragile X patients suffer from seizures, <https://doi.org/10.3389/fped.2021.736255>; 12-27% of SHANK3 subjects have epilepsy, PMID: 27554343; 11% in 22q11 PMID: 30977115. This inference is also present in the discussion (page 17 line 448). An alternative conclusion is that the authors have identified targets and pathways relevant to the epilepsy/seizure phenotypes, and these may or may not be independent from SZ and ASD phenotypes. The balance of data in the paper – specifically, one behavioral test in mice with minimal face validity – does not support the expansive conclusions in the paper. The overlap with SZ and ASD postmortem tissue is again not conclusive, given the comorbidity across these disorders. There is nothing here to distinguish whether the protein network the authors have discovered is specific to seizures or relevant to a wider array of NPD phenotypes.

The reviewer has identified an important caveat in this analysis that was not discussed in the text. Given the prevalence of seizures in many genetic forms of NPDs we cannot emphatically distinguish whether the proteomic network alterations are convergent across many NPDs or linked specifically to the seizure aspect of each disorder (i.e. co-morbid epilepsy). Based on the reviewer’s suggestion, we have therefore moved this panel to supplementary Fig 3 and re-interpreted this analysis, removing any expansive conclusions in the text

Based on the reviewer's comments, we have also rewritten the introduction to focus on epilepsy as a co-morbid phenotype.

Please refer to the new introduction, and observe less expansive text relating to the new supplementary Fig 3A below:

Results

"Together these data indicate the potential for shared proteomic pathophysiology across different genetic mouse models with diverse etiologies, and reveal that elements of the epilepsy network are disrupted in genetic mouse models and human NPDs"

Discussion

"These data may be revealing of a molecular signature associated with NPDs and co-morbid epilepsy."

Page 12, line 311: This paragraph is somewhat confusing. The authors identify membrane proteins that are changed when the entire 16p11 region is duplicated. They then identify proteins that interact with a single gene in the region, PRRT2. It seems that overlapping targets contained in these two groups is to be expected? Why would this qualify as "surprising overlap" (line 319)? If PRRT2 is disrupted in the 16p11.2 duplication, wouldn't one expect this to encompass the entire PRRT2 proteome? The fact that only 72% of the PRRT2 interactome is disrupted in the larger 16p11 dup proteome seems to be less than what is expected, not more. I would want to know the numbers in the other direction: how much of the entire 16p11 dup disrupted proteome is accounted for by duplication of PRRT2 alone? Those data might demonstrate that PRRT2 is in fact a driver of the underlying biological perturbations.

Thank you for your comments. We initially found this overlap surprisingly high because we felt that a 100% overlap would only occur in a static system with perfect experimental conditions. However, we always expect a degree of biological noise in omic experiments and compensatory mechanisms are known to occur within neuronal membranes (homeostatic plasticity). Thus, even though PRRT2 is known to regulate membrane trafficking, we were impressed the overlap was so high.

We do however see that the word "surprisingly" is very subjective and depends on perspective. We have therefore removed the use of this word and provided additional data on the proportion of the disrupted proteome which interacts with PRRT2 (~8%), which helps interpret our data further. If each gene within the 16p11.2 were to contribute to membrane changes this would represent a 3.7% (1/27) change for each, thus PRRT2 contributes to more than twice what may be expected in this scenario.

"Remarkably, we show that PRRT2-interacting proteins were systematically disrupted in the membrane proteome of 16p11.2^{dup/+} mice (72% of interacting proteins), and conversely, 8.3% of the disrupted membrane proteome were part of PRRT2-complexes (Fig. 4G)."

Page 14, line 382: correction of the induced seizure phenotype is a very strong finding and the most rigorous evidence in the paper that PRRT2 may drive seizure phenotypes. However, the

reversal of the social deficit phenotype, while encouraging, cannot reasonably be interpreted as a marker of all NPD phenotypes, nor can it be interpreted as a “core behavioral endophenotype associated with ASD and SZ.” Other phenotypes associated with SZ and ASD have not been tested such as marble burying (stereotypy), vocalization frequency (deficits in language), and PPI (especially relevant for SZ). Furthermore, no learning and memory phenotypes were tested. I do not believe that this paper conclusively demonstrates that “correction of PRRT2 dosage can rescue core neuropsychiatric domains” in 16p11.2 duplication mice.

We do apologize for the lack of clarity in our writing. We agree that the social phenotype is not a “marker of all NPDs” or a “core behavioural endophenotype”, and based on the reviewer’s comments, we have retracted those sentences and removed any expansive interpretations from the text. We also concur that there are many other behavioural phenotypes relevant to NPDs that we could have measured.

However, we would like to indicate that social phenotypes are present in many NPD-relevant mouse models, and have been categorized as a core symptom domain in the NIMH RDoC initiative (Social Processes). Thus, we wanted to use this behaviour as NPD-relevant phenotype not directly linked to epilepsy and seizures, and show that we could rescue a non-epileptic phenotype. We believe this extends the importance of our findings and remains an important discovery.

We have therefore substantially edited the text and simplified our conclusions, such as:

Discussion

“Here, we have shown that reversing circuit phenotypes in the *16p11.2^{dup/+}* mice is also associated with a rescue of social deficits, a non-epileptic phenotype. However, the relevance of the circuit phenotype to the wide variety of symptoms exhibited in human carriers, as well as the exact brain regions and circuits involved, remains to be fully determined.”

It is somewhat surprising that the authors compared their proteomic data to other mouse models and human datasets, but do not present the overlap between the 16p11.2 dup dysregulated proteome and the transcriptome, of which there are many available datasets.

Thank you for this suggestion. We now provide an in-depth analysis of RNA-protein correlations using RNA-seq data from two publications using the same mouse model as us (Fig. S2). This analysis revealed a poor level of correlation between the RNA and protein (outside the duplicated region), suggesting that most of the membrane changes we observe are transcription-independent, and likely due to trafficking or proteostasis. We include the following text in the manuscript:

Results

“Alterations in membrane protein abundance could be caused by several mechanisms including transcriptional dysregulation, proteostasis and protein trafficking. To gain insight into the molecular mechanisms involved, we compared our proteomic data to two published datasets that used RNA-seq to profile gene expression changes in the whole neocortex or prefrontal cortex of *16p11.2^{dup/+}* mice^{30,38} (Fig. S2A-C). In the whole neocortex dataset, we found 41 genes that overlapped with the membrane proteome changes (Fig. S2A). Gene set enrichment analysis revealed this overlap was not statistically significant, both with, or without 16p11.2 genes included

in the analysis (Fig. S2D-E). We also considered genes/proteins in the whole cortex dataset that were altered in the same direction (i.e. up in RNA-seq vs up in membrane proteome or down in RNA-seq and down in membrane proteome), and again found no statistically significant enrichment (Fig. S2D-E). We next performed a correlation analysis with gene/proteins in the overlap (n=41) and found that they were significantly correlated ($p = 0.0378$), albeit with a poor linear relationship ($R^2=0.022$) (Fig. S2F). However, the significance of this correlation disappeared ($p = 0.1638$) when genes in the 16p11.2 locus were removed, indicating that the correlation was dependent on cis effects of the 16p11.2 genes (Fig. S2G). In the prefrontal cortex dataset, we found 53 genes that overlapped with the membrane proteome changes, which represented an enrichment more than was expected by chance ($p = 0.0036$) (Fig. S2B and S2H). This enrichment persisted when we removed the 16p11.2 locus genes ($p = 0.0024$) (Fig. S2I). However, when considering genes/proteins that were altered in the same direction, we found no statically significant enrichment, which was corroborated by the lack of correlation between the two data sets (Fig. S2J-K). Together these data indicate a lack of correlation between gene expression and membrane proteomic changes in for genes/proteins located outside of the 16p11.2 locus, and suggest that membrane changes are likely not due to global transcriptional effects.”

Discussion

“To assess the level of overlap between RNA and protein, we compared our membrane proteome data to previously published RNA-seq datasets in the same mouse model. We found little to no correlation between changes in mRNA and protein exemplifying the importance of performing proteomic analysis in addition to transcriptomics, to fully capture the disease process. The lack of correlation between RNA and proteins suggests that although transcriptional effects may explain some aspects of disease, they cannot explain how membrane proteins are altered in this model. It is therefore likely that post-transcriptional effects regulating protein abundance and localization are involved. Given that PRRT2 has been show to regulate surface trafficking of membrane proteins, and that it interacts with an abundance of proteins that are dysregulated, we would propose that PRRT2-dependent membrane trafficking could be an important player in remodeling the membrane proteome of *16p11.2^{dup/+}* mice.”

Reviewer #2 (Remarks to the Author):

Human chromosome 16p11.2 is one of the recurrent CNV loci, including deletion and duplication in ASD, SZ, or ID individuals. The authors conducted quantitative proteomics analysis followed by bioinformatics approaches to identify a driver gene or dysregulated network in the brain of the 16p11.2 duplication mouse model. They found that an epilepsy-associated protein subnetwork was especially dysregulated in 16p11.2dup/+ mice. In addition, calcium imaging analyses by using primary cortical neurons or slices derived from 16p11.2 dup/+ mice showed enhanced functional connectivity or glutamate release. Next, the authors focused on the Prrt2 gene, which is likely to be the most reasonable gene for pathogenesis of 16p11.2 duplication based on their bioinformatics analyses. They performed the Prrt2 interactome and found that all known proteins that bind to Prrt2 are listed in the candidates. These candidates are again enriched in the epilepsy network, suggesting Prrt2 may have a central role in the epilepsy network in 16p11.2 genes. Finally, the authors corrected the copy number of Prrt2 in 16p11.2 by crossing with heterozygote of Prrt2 mice, named 16p11.2corr. This line showed a normalized level of seizure susceptibility or abnormal social interaction found in 16p11.2 dup/+ mice. Taken together, the authors concluded that the proteomics approach helps identify not only genes but also disease hubs in complex genetic disorders such as CNVs.

The authors' analyses or strategies are sophisticated and quantitative. So, the results seem to be reliable and convincing conclusions. The reviewer has very few comments.

#1. Page 8, Fig. 2A-B: From this sentence, we cannot know whether these samples are primary neurons, slice cultured neurons, or in vivo neurons.

The type of sample used has now been clarified in the Figure legends, text subheadings, and description of the results

#2. Page 21: The genetic background of 16p11.2 dup/+ mice should be described clearly. After obtaining this line, how many times did the authors cross with C57BL/6N?

We apologize for omitting this information. It has now been added to the methods sections.

"Male 16p11.2^{dup/+} mice were then crossed for at least 2 generations on a C57BL/6N background (Charles River Laboratories) prior to experiments"

#3. As the authors previously showed, 16p11.2 dup/+ mice exhibited increased dendritic arborization (K D Blizinsky et al., 2016). This is an interesting view to see dendric arborization in 16q11.2Corr mice.

Thank you for your interest in our previous work. We have actually examined dendritic phenotypes in neurons from Dup and Corr mice. [REDACTED]

[REDACTED]

Reviewer #3 (Remarks to the Author):

Here the authors describe a mouse model of CNV previously associated with various neurological disorders, the microduplication 16p11.2. Using this mouse model, they performed unbiased proteomic assays on membrane proteins in brain and identified a large amount of proteomic dysregulation, including networks involved in epilepsy, which were also associated with hypersynchronous activity in neurons and slices from 16p11.2 mutant mice. They also performed rescue experiments using a single gene, Prrt2, to reverse electrophysiological and behavioral, namely seizures, phenotypes. Overall, the manuscript was well-written and included important data and mechanistic insights into the seizure phenotype associated with this CNV. However, some major questions were left unanswered and the framing of the paper could be substantially improved.

Positive comments:

1. Use of unbiased proteomics on membrane proteins in this CNV mouse model is an important dataset for the field, especially because this CNV is associated with so many different phenotypes and disease outcomes in humans.
2. Calcium imaging experiments and behavioral phenotyping for seizures, including rescue of the phenotype with Prrt2, were fascinating and will be of great interest to the epilepsy/seizure field.

Major drawbacks:

1. The way the authors frame the manuscript, as this CNV being associated with shared risk for neuropsychiatric disease, especially ASD and SZ, seems out of place. Primarily, these two disorders seem cherry picked from the list of diseases associated with 16p11.2 microduplication, but then the entire paper focuses on epilepsy and seizure phenotypes, which is arguably a neurological disorder but not a neuropsychiatric disorder. The reviewer recommends the authors focus on the epilepsy phenotypes and not cloud the manuscript with less relevant information about shared risk with psychiatric disease, as their imaging and behavioral data only support the former. Further, Figure 1 shows various datasets the authors mined to create a link between their proteomics data and various neuropsychiatric/neurological diseases, and this seems unnecessary based on the later experiments.

We thank the reviewer for their perspective and do agree that some aspects of the work may have more obvious implications for epilepsy/seizure risk. However, we feel we would be remiss if we focused the manuscript entirely on epilepsy, without considering the full effects of this CNVs on disease risk. Our reasoning is as follows:

- 1- We would argue that the only phenotype we assess which is exclusively relevant to epilepsy risk is the seizure susceptibility assay. We do emphasize that there are no spontaneous seizures in this model, thus the susceptibility to kainate is likely revealing an underlying “subclinical” circuit dysfunction, rather than an overt model for epilepsy. The hypersynchrony phenotype can, however, be interpreted in many ways (see point 2).
- 2- The effects of this CNV are highly pleiotropic, causing a wide range of clinical diagnoses in humans ranging from epilepsy to schizophrenia, autism, and ADHD. Therefore, any pathological processes we identify in the 16p11.2 mouse model could be linked to one or several behavioural phenotypes. For example, the changes in circuit function

(hypersynchrony) could have a multitude of effects on cortical processing and thus impact distinct behavioural domains, which is one of the reasons we find this result to be meaningful. Although hypersynchrony is most commonly associated with electrographic seizures in the medical field, a large body of evidence in the basic and cognitive neuroscience field has shown that synchrony regulation is critical for processing information in the brain, and is also altered in neuropsychiatric disorders (distinct from epilepsy) [PMID: 17015233, 25201983, 32956387]. Thus, we believe this circuit phenotype can be interpreted in both ways and it is indeed important to do so given that the 16p11.2 duplication produces both epilepsy and psychiatric disorders in humans. Our rescue of social behaviour would in fact support this broader interpretation of the results too.

- 3- We do like to use the term neuropsychiatric to refer broadly to both psychiatric and neurological disorders. We feel this is especially important when considering CNV disorders which are at the interface of these disciplines. It must be noted that epilepsy in general, has high rates of psychiatric diagnoses (up to 56%, PMID: 23021287) making it a neuropsychiatric disorder by broader definitions. Thus, it is increasingly being recognized that the distinction between the fields of psychiatry and neurology is largely historical and not necessarily biological [for a great discussion please see 11986119].

We, therefore, propose the following modifications to better balance the text in light of the reviewer's concerns:

- i) We have focused our introduction on epilepsy as a co-morbid phenotype in NPDs as a whole, as this puts a fair accent on epilepsy without excluding the wider phenotypic spectrum of the duplication syndrome. We do reference the literature for ASD and SZ in some areas because these are the most statistically robust and reproducible genetic associations to date. Other genetic associations such as ADHD, bipolar and Rolandic epilepsy will still require validation in larger cohorts.
- ii) We have modified our discussion to more clearly separate the different implications of the circuit phenotype. Please see "Implications of circuit phenotypes for epilepsy" and "Implications of circuit phenotypes for other NPDs" in the discussion section.
- iii) The proteomics data in Figure 1 has been moved to supplementary to de-emphasize these findings, and have now been re-interpreted in light of reviewer concerns. We do acknowledge that co-morbid epilepsy could be a factor in explaining the shared molecular disruptions (put forward by Reviewer 1).

2. Similarly, the inclusion of social behavior data in the last figure seems extremely out of place as well, as none of the mechanistic studies (which are very interesting in the context of seizures and epilepsy) can be directly related to social behavior, and these tests should not be included.

We do apologize for the lack of clarity in our manuscript. We hope that point 2 above has explained this point in more depth. We do believe it was very revealing to see that restoration network function is accompanied by improved social behaviour. These data suggest that circuit hypersynchrony may in fact underlie both seizures and social dysfunction. As disrupted synchronization in cortical circuits has been linked to social impairments [PMID: 34433814, 29503190], we believe that the rescue of this non-epileptic phenotype remains an important

discovery, which we should keep in the manuscript. These data may aid future studies to further clarify the precise mechanisms of social deficits in this model.

Additionally, in line 77, the authors state that deficits in social behavior exhibited by 16p11.2 mice are common to SZ and ASD, and this is quite a stretch, as social deficits are involved in most psychiatric diseases and only a core symptom of ASD, not SZ.

We agree with this reviewer that social deficits are an important component of most psychiatric diseases (not just ASD and SZ). This is indeed one of the reasons why we chose to look at this phenotype. Based on the reviewer's concern, we removed this phrase from the new introduction. We also show restraint in extrapolating mouse behaviours to human phenotypes. Please see the relevant new text in the discussion:

"Here, we have shown that reversing circuit phenotypes in the *16p11.2^{dup/+}* mice is also associated with a rescue of social deficits, a non-epileptic phenotype. However, the relevance of the circuit phenotype to the wide variety of symptoms exhibited in human carriers, as well as the exact brain regions and circuits involved, remains to be fully determined."

"Despite this, care must be taken to draw parallels between human and mouse phenotypes. The *16p11.2^{dup/+}* mouse model has strong construct validity because the same set of genes are duplicated in mice compared to human conditions. However, it is not possible to discern whether behavioral deficits exhibited in mice (face validity) are a true reflection of social impairments observed in NPDs. Sociability alterations in *16p11.2^{dup/+}* mice could represent phenocopies of human symptoms that are not mechanistically linked⁷⁴. Nevertheless, social deficits appear to be one of the most reproducible behavioral phenotypes in *16p11.2^{dup/+}* mice and are rescued correcting PRRT2 expression"

3. The authors should include an additional experiment investigating the transcriptome of these mice using RNA-seq. They state in the manuscript that transcriptomic outcomes are important for CNVs and this one in particular, yet do not include that data and jump straight to proteomics.

We do confirm that RNA-seq can be a useful approach to unravel mechanisms in CNV mouse models. In the case of *16p11.2^{dup/+}* mice, it has already been performed by a number of groups (two published and one in biorxiv, <https://doi.org/10.1101/2022.05.12.491670>). For this reason, we chose to focus our study on proteomics as a complementary approach.

To address this concern, we have included a comprehensive analysis comparing two published RNA-seq datasets to our proteomic data. This analysis revealed a poor level of correlation between the RNA and protein (outside the duplicated region), suggesting most of the membrane changes we observe are transcription-independent, and likely due to proteostasis or trafficking. We include the following text in the manuscript:

Results:

Alterations in membrane protein abundance could be caused by several mechanisms including transcriptional dysregulation, proteostasis and protein trafficking. To gain insight into the molecular mechanisms involved, we compared our proteomic data to two published datasets that used RNA-seq to profile gene expression changes in the whole neocortex or prefrontal cortex of

16p11.2^{dup/+} mice^{30,38} (Fig. S2A-C). In the whole neocortex dataset, we found 41 genes that overlapped with the membrane proteome changes (Fig. S2A). Gene set enrichment analysis revealed this overlap was not statistically significant, both with, or without 16p11.2 genes included in the analysis (Fig. S2D-E). We also considered genes/proteins in the whole cortex dataset that were altered in the same direction (i.e. up in RNA-seq vs up in membrane proteome or down in RNA-seq and down in membrane proteome), and again found no statistically significant enrichment (Fig. S2D-E). We next performed a correlation analysis with gene/proteins in the overlap (n=41) and found that they were significantly correlated ($p = 0.378$), albeit with a poor linear relationship ($R^2=0.022$) (Fig. S2F). However, the significance of this correlation disappeared ($p = 0.1638$) when genes in the 16p11.2 locus were removed, indicating that the correlation was dependent on cis effects of the 16p11.2 genes (Fig. S2G). In the prefrontal cortex dataset, we found 53 genes that overlapped with the membrane proteome changes, which represented an enrichment more than was expected by chance ($p = 0.0036$) (Fig. S2B and S2H). This enrichment persisted when we removed the 16p11.2 locus genes ($p = 0.0024$) (Fig. S2I). However, when considering genes/proteins that were altered in the same direction, we found no statically significant enrichment, which was corroborated by the lack of correlation between the two data sets (Fig. S2J-K). Together these data indicate a lack of correlation between gene expression and membrane proteomic changes in for genes/proteins located outside of the 16p11.2 locus, and suggest that membrane changes are likely not due to global transcriptional effects.

Discussion:

To assess the level of overlap between RNA and protein, we compared our membrane proteome data to previously published RNA-seq datasets in the same mouse model. We found little to no correlation between changes in mRNA and protein exemplifying the importance of performing proteomic analysis in addition to transcriptomics, to fully capture the disease process. The lack of correlation between RNA and proteins suggests that although transcriptional effects may explain some aspects of disease, they cannot explain how membrane proteins are altered in this model. It is therefore likely that post-transcriptional effects regulating protein abundance and localization are involved. Given that PRRT2 has been show to regulate surface trafficking of membrane proteins, and that it interacts with an abundance of proteins that are dysregulated, we would propose that PRRT2-dependent membrane trafficking could be an important player in remodeling the membrane proteome of 16p11.2^{dup/+} mice.

4. The above data (transcriptomics) would also be useful in the case of explaining the large number of proteomic changes reported that are NOT directly attributable to the genes in the genomic region altered by this CNV. This should be further explained and investigated in the manuscript, as ~1600 differentially expressed proteins is very different from the 27 protein coding genes involved in that genomic locus.

This is an excellent point that we have not addressed in the text. The additional insight from the transcriptomic datasets has allowed us to infer how the protein alterations may be occurring. The proposed mechanisms have now been added to the discussion (see above) and explained further in the figure below (Fig. 2). Based on our findings and previous publications, our hypothesis is that PRRT2 functions as a trafficking adaptor in synaptic vesicles allowing it to alter the membrane levels of interacting proteins, in a transcription-independent manner. Thus, although only one

protein (PRRT2) is being overexpressed, it can alter membrane levels of hundreds of proteins, reshaping the composition of neuronal membranes in *16p11.2^{dup/+}* mice.

Fig 2. Potential mechanism of membrane proteome dysregulation in *16p11.2* dup mouse model. Some genes have elevated mRNA and increased membrane levels (Class I). Class one genes are mainly related to the *16p11.2* locus (e.g. PRRT2 and SEZ6L2). There is an increase in mRNA from the gene duplication, a proportional increase in protein, and thus, more is found at the membrane. However, some genes have no change in mRNA but have changes in membrane levels (Class II). These tend to be the ones interacting with PRRT2. Hence, we propose that Class II changes occur by altered trafficking to the membrane via PRRT2-dependent mechanisms.

5. There is no evidence of effect size or subject/cell numbers (N) in any of the manuscript or figures, and this is especially important in experiments where it is unclear if the data points on the graph represent one biological replicate (animal) or one cell (for example, all calcium imaging experiments)

We have modified all figure legends to include the numbers (N) for each experiment and what type of sample they represent. This information can also be found in the methods section.

6. The authors show an "increased burden of risk factors associated with many *16p11.2* related NPDs" in Fig 1 based on their proteomic data, but the data appears to basically include a large number of proteins involved in synaptic functioning that are generally impaired when brain disease is present. It may be better to focus on their neural transmission role and not necessarily attribute this to a specific disorder.

Synapse dysfunction is certainly a convergence point for many NPDs, however, we did find it notable that when we compared proteomic dysregulation in the mouse model to genetic variants identified in NPDs, there was a clear difference in the enrichment between diseases. This indicates that not all disease networks are composed of the same genes/proteins and that the type of disease enrichment may be useful in predicting downstream phenotypes, as has been performed in this study (enrichment of epilepsy genes and phenotype of seizure susceptibility). We aim to further investigate the role of the specific synaptic pathways in future work.

Other comments:

1. Why do the authors choose membrane proteins to screen compared to other subcellular fractions or total cell lysate? Can the authors clarify if the membrane fractions are essentially synaptosomes or if they are distinct in some way?

The biochemical preparation we used for the MS is called a P2 pellet or membrane fraction. It is also sometimes called a crude synaptosomal fraction, as it is essentially the first step towards making pure synaptosomes (which requires an additional centrifugation and extraction step). The P2 fraction is therefore enriched in synaptosomes but will also contain other plasma membrane proteins, and large organelles such as lysosomes and mitochondria.

We have added a line in the methods section to clarify this:

"We next prepared P2 membrane fractions (or crude synaptosomes) from the frozen cortical tissue."

We chose to use membrane fractions rather than whole tissue lysates, because changes in lower abundance proteins that reside in the membrane may be lost when doing whole tissue proteomics. Although mass spectrometry is currently the best available technique for unbiased proteomic screens, it still has limited dynamic range compared to RNA-seq for instance. A high dynamic range is critical to identify less abundant peptides in sample with more abundant ones [PMID: 21616670]. Thus, using whole tissue lysates may mask a lot of important changes at the membrane due to an excess of abundant cytosolic, nuclear or ribosomal proteins in the sample. Lowering sample complexity (i.e. extracting a particular biochemical fraction) can therefore reveal more specific proteomic changes in compartments of interest such as the membranes, a key site of cell-cell communication in the brain.

2. Line 179- membrane protein or total protein in referenced study?

It was in fact the membrane protein alterations. Thank you, this has been corrected

3. In figure 2, it is not clear right away that these studies are done in neuronal cultures and not in vivo due to the wording of the figure caption and manuscript. It would be useful to distinguish this and to also distinguish it from the following data using slices instead of cells.

This is an important distinction. It has been clarified in the Figure legends, subheadings and text of the results.

4. In Figure 2 A-B, a wild type comparison figure is needed.

Thank you, this has now been added to Figure 2

5. Figure 6D, F- need legend to denote groups for each color on these graphs.
We have added a colour key to Figure 6

Reviewer #4 (Remarks to the Author):

A major problem in interpreting deletions and duplication of multigenic regions is untangling the role that each affected gene plays in producing the phenotype. There are many CNVs that produce neurological disorders and the 16p11.2dup/+ mice are an ideal model system. The authors have done an outstanding job in unravelling how the duplicated region produces complex phenotypes through the integration of many methods. The project exemplifies how proteomic approaches that leverage the wealth of publicly available bioinformatics can lead to the hypothesis that PRRT2 overexpression is a key cause of the phenotypes in 16p11.2dup/+ mice, which they substantiate using a genetic rescue approach. I recommend publishing the manuscript and suggest the following modifications to the text:

On page 6, the authors perform “gene set analysis to determine if the dysregulated proteins contained a burden of disease associated gene products identified by de novo exome sequencing studies of NPDs”. They say “We found an increased burden of risk factors associated with many 16p11.2dup/+ related NPDs in our dataset (Fig. S1G and Fig. 1D). These data are reminiscent of the diverse clinical spectrum observed in CNV carriers and suggest that subnetworks relevant to each disease may be dysregulated in 16p11.2dup/+ mice.

”How do we know this doesn’t just reflect an enrichment in the expression of NPDs in the brain? i.e. neural expressed proteins are going to be enriched in neural gene mutations. For this reason I do not agree that the following statement is surprising: “Surprisingly, the most enriched disease gene set was associated with epilepsy risk factors, followed by ID, SZ and ASD gene sets”. The authors need to explain in the results section how they account for the bias in enrichment that results from brain gene expression. If they cannot do this, they need to tone down the text. I accept that they can present the data to make the link to epilepsy.

We thank the reviewer for raising this important point, which we did not explain clearly enough in the results section. Indeed, finding an enrichment of NPD-associated gene variants in a list of proteins that are brain expressed would not be that surprising. To correct for the bias of brain expression and membrane localization (as may NPD risk genes are known to be synaptic), we used the total proteome detected in our membrane fraction as the background for all of our analyses. Thus, we only report enrichments in the dysregulated protein datasets that are beyond what would be expected from the membrane background. We have included a sentence in the text to clarify this point. This initial description of this analysis was included in the methods but we accept that this may not be easy to find.

“Importantly, we used all identified membrane proteins within our dataset as a background for these analyses, removing any bias which may arise from brain expression or membrane localization.”

Page 6, last paragraph, which reads “To determine the clinical relevance of these molecular disease-associations...” The authors should explain how they calculate the conferred risk. I also think it is very ambiguous to say “clinical relevance” because what is clinically relevant is to most doctors is how they treat patients. It would be more informative to give a more accurate description that aligns with the analysis they present, which appears to be a ranking of diseases.

We agree that the word clinically relevant may not be appropriate in this context. As this dataset was problematic for another reviewer (previously Fig 1E), the figure panel and associated text has now been removed. Other instances referring to “clinical relevance” have also been edited.

Page 7, the second paragraph says: “To determine the broader relevance of this network, and the global proteomic changes in 16p11.2dup/+ mice, we compared our datasets to disrupted proteins in a range of NPD mouse models, and human post-mortem proteomic studies of ASD and SZ (Fig. 1I). Gene set analysis revealed that a majority of mouse models we evaluated were enriched for global protein alterations observed in 16p11.2dup/+ mice (Fmr1-/y, Shank3-/-, 22q11.2+/-, Cul3-/-). Human datasets for ASD and SZ were remarkably, also enriched, indicating a convergence of human and mouse disease processes.”

What does it mean to say “we compared our datasets to disrupted proteins in a range of NPD mouse models, and human post-mortem proteomic studies of ASD and SZ” and “Human datasets for ASD and SZ were remarkably, also enriched”? For the reader to understand this they need to dig around in some supplementary information or methods section (without being directed there by the authors) and even then, it is not obvious. This paragraph is vague and needs a better description.

This section has been edited to more carefully explain the analysis. We have also separated the gene set enrichment analysis section (which was very long) into two parts (“Gene set enrichment analysis” and “Custom gene lists”), so that the method details can more easily be found. Please find the edited results section below:

“To determine the broader relevance of this network, we aimed to determine whether proteins disrupted in the epilepsy network and membrane proteome of 16p11.2^{dup/+} mice were also altered in NPD mouse models, and human post-mortem brain tissue in ASD and SZ (Fig. S3D). To this end, we overlapped lists of proteins dysregulated in the 16p11.2^{dup/+} mouse model with dysregulated proteins from mouse or human datasets, and assessed the extent of overlap between with each list. Gene enrichment set analysis was then used to determine whether the overlap was more than was expected by chance. We found that...”

Page 8, Section entitled “Enhanced functional connectivity etc”. There is no mention in this paragraph of whether they were looking at neurons in the intact mouse or cultures (the reader has to go to the figure legend). Again, a problem with clarity of description.

Thank you for highlighting this. We have specified the type of sample we are analysing more clearly in the Figure legends, text subheadings, and description of the results.

Figure 1B. chromosome is misspelled.

Thank you, this has been corrected.

Seth Grant
Edinburgh University

REVIEWER COMMENTS

Reviewer #2 (Remarks to the Author):

The authors have responded to my comments.

Reviewer #3 (Remarks to the Author):

The authors have addressed all of my comments sufficiently. I especially find the manuscript much improved with the inclusion of previously published transcriptomic datasets that have now been compared to the current proteomic data.